# The extracellular matrix regulates cortical layer dynamics and cross-columnar frequency integration in the auditory cortex

Mohamed El-Tabbal [1,5✉], Hartmut Niekisch[1,6], Julia U. Henschke[2], Eike Budinger [1,3], Renato Frischknecht [4], Matthias Deliano[1,7] & Max F. K. Happel [1,3,7✉]

In the adult vertebrate brain, enzymatic removal of the extracellular matrix (ECM) is increasingly recognized to promote learning, memory recall, and restorative plasticity. The impact of the ECM on translaminar dynamics during cortical circuit processing is still not understood. Here, we removed the ECM in the primary auditory cortex (ACx) of adult Mongolian gerbils using local injections of hyaluronidase (HYase). Using laminar current-source density (CSD) analysis, we found layer-specific changes of the spatiotemporal synaptic patterns with increased cross-columnar integration and simultaneous weakening of early local sensory input processing within infragranular layers Vb. These changes had an oscillatory fingerprint within beta-band (25–36 Hz) selectively within infragranular layers Vb. To understand the laminar interaction dynamics after ECM digestion, we used time-domain conditional Granger causality (GC) measures to identify the increased drive of supragranular layers towards deeper infragranular layers. These results showed that ECM degradation altered translaminar cortical network dynamics with a stronger supragranular lead of the columnar response profile.

[1] Department of Systems Physiology of Learning, Leibniz Institute for Neurobiology, 39118 Magdeburg, Germany. [2] Institute of Cognitive Neurology and Dementia Research (IKND), Otto von Guericke University, Magdeburg, Germany. [3] Center for Behavioral Brain Sciences (CBBS), 39106 Magdeburg, Germany. [4] FAU Erlangen-Nürnberg, Animal Physiology, Department of Biology, 91058 Erlangen, Germany. [5] Present address: Optical neuroimaging unit, Okinawa Institute of Science and Technology, Okinawa, Japan. [6] Present address: Department of Biology, Animal Physiology, Technical University, Kaiserslautern, Germany. [7] These authors contributed equally: Matthias Deliano, Max F. K. Happel. ✉email: mohamed.eltabbal@oist.jp; mhappel@lin-magdeburg.de

While basic neuronal networks established during development must be conserved, their activity-dependent fine-tuning and modification form the basic mechanism for adult learning and memory. In the vertebrate brain, long-lasting structural tenacity of neuronal networks is supported by the extracellular matrix (ECM). It consists of the core backbone glycan hyaluronic acid that attaches chondroitin sulfate proteoglycans and other proteins[1,2]. The most prominent forms of the adult ECM in the brain are (i) perineuronal nets (PNN) named by their mesh-like appearance around cell bodies and proximal dendrites of mainly parvalbumin-positive (PV + ) interneurons, and (ii) the "loose" ECM found almost ubiquitously in the brain[2,3]. The ECM is formed during brain maturation implementing a switch from juvenile to adult forms of synaptic plasticity[4,5]. For instance, enriched PNNs have been linked to states of reduced structural plasticity[6,7]. In a seminal study, Pizzorusso and colleagues demonstrated, that enzymatic removal using chondroitinase ABC reinstalls critical period plasticity[4]. ECM removal within the cortex or hippocampus has further diverse effects on contextual fear conditioning[8,9], object recognition[10], extinction of fear or drug memories[11,12], and auditory reversal learning[13]. During learning processes in the developing and adult brain, variability of ECM densities and dynamic remodeling has been shown to support learning-dependent plasticity[14,15]. In accordance, systemic injection of dopamine receptor D1 agonists in vivo promote rapid cleavage of the ECM protein brevican[16]. ECM weakening increases lateral diffusion of glutamate receptors at the synapse[17–20] and alters the balance between inhibition and excitation[21]. However, the mechanistic understanding on how states of reduced ECM may affect circuit processing in a layered structure, such as the cortex, is rather elusive.

Here, we acutely weakened the ECM in the primary auditory cortex (ACx) of adult Mongolian gerbils (*Meriones unguiculatus*) by local microinjections of hyaluronidase (HYase). In parallel, we used laminar recordings of local field potentials and current-source density (CSD) analysis[22,23] to quantify the spatiotemporal sequence of spontaneous and stimulus-evoked laminar synaptic activation. Local HYase injection led to an imbalance between the synaptic activity in supragranular layers I/II and infragranular layer Vb. The imbalance was mediated by a moderate increase in tone-evoked synaptic currents in supragranular layers I/II and a stronger stimulus-specific decrease in infragranular layer Vb. These changes in the spatiotemporal synaptic pattern were accompanied by an increase of lateral corticocortical input revealed by CSD residual analysis[22]. Utilizing a multitaper spectral analysis of tone-evoked oscillatory circuit responses[24], we revealed a layer specific change of the oscillatory power in the beta band (25–36 Hz) within infragranular layers Vb. To investigate how these synaptic spatiotemporal changes are mediated by the translaminar dynamics of cortical network activity, we implemented time-domain Granger causality (GC) measures mapping the directional causal relationship between activity across cortical layers. After application of HYase, we found a significant increase in the GC from supragranular layers I/II onto infragranular layer VI. Additionally, there was a significant increase in the infragranular layer VI drive routing towards early infragranular layers Vb which decreased its drive towards granular layers III/IV. Our data thereby shows that enzymatic removal of the ECM acutely biases the columnar synaptic network processing towards stronger recruitment of supragranular circuits and enhancement of lateral, cross-columnar interactions along with stronger drive from supragranular layers I/II towards deep infragranular layers VI which also showed stronger drive towards layer Vb. Our study unraveled a mesoscopic cortical circuit mechanism of enhanced sensory integration in upper layers controlled by the ECM. These findings help to better understand existing behavioral findings linked to ECM modulation and may further allow to better understand certain pathological conditions of altered translaminar response characteristics, as for instance during sensory loss, and may help to optimize concepts for therapeutic approaches targeting the ECM.

## Results

In the present study, the extracellular matrix was locally removed in unilateral primary auditory cortex of adult Mongolian gerbils using microinjection of the ECM-degrading enzyme hyaluronidase. By placing a linear multichannel recording electrode in the near proximity (<250 μm) of the glass capillary, we recorded the laminar local field potential across all layers within the region of the significantly reduced ECM. CSD analysis was used to investigate how ECM removal acutely affects laminar tone-evoked processing and spontaneous activity. Specifically, analysis of the relative residual of the CSD, layer-specific multitaper spectral analysis, and Granger causality measures allowed us to identify enhanced corticocortical input in upper layers to dominate the translaminar processing dynamics within the cortical column[22,23].

**Layer-dependent changes of tone-evoked synaptic activity**. Pure tone-evoked laminar current source density profiles were recorded in the untreated primary auditory cortex (see Eq. 1 in Methods and materials). Current sinks thereby reflect the spatiotemporal cascade of excitatory synaptic population activity across cortical layers, while current sources mainly reflect passive return currents (further information see Methods and materials)[22,25]. Recordings showed a canonical feedforward processing pattern with main short-latent current sink components in granular layers III/IV and infragranular layer Vb most prominently for stimulation with the best frequency (BF; Fig. 1a, *left*). Those short-latent sinks reflect the synaptic afferent input from the lemniscal auditory part of the thalamus, the ventral medial geniculate body (vMGB), with main projections to granular layers and collaterals within layer Vb[26–28]. This has been revealed by earlier studies showing persistent thalamocortical inputs in corresponding layers after intracortical silencing with the GABA$_A$-agonist muscimol[22,23,29] in accordance with reports by others[28,30]. Early sinks were followed by synaptic activation of supragranular (I/II) and deep infragranular (VI) layers (Fig. 1a), which are due to intracortical synaptic processes[22]. Then, the ECM-degrading enzyme HYase was injected in close proximity at the recording site (the micropipette was inserted ~250 μm next to the electrode). Before the start of the recording and tissue was allowed to recover from insertion for >1 h. After HYase injection we again waited for >2.5 h in order to let the enzyme effectuate the ECM in the proximity of the recording patch. Stability of the recording track along the cortical laminae over time has been controlled by a statistical correlation method, which assumes that the general profile of sinks and sources across recording channels should be stable irrespective of relative amplitude changes. Onset latencies within layers receiving short-latent thalamocortical input were stable across both conditions (Fig. 1b). Hence, we channel-wise averaged the first 50 ms from the stimulus onset in the CSD profile and cross-correlated the resulting 32-channels of before and after enzyme injection along the derivation axis with a shift in space. We found peaks of this correlation to be present at $0 \pm 1$ channel shift, which corresponds to a shift of maximally 50 μm within all individual

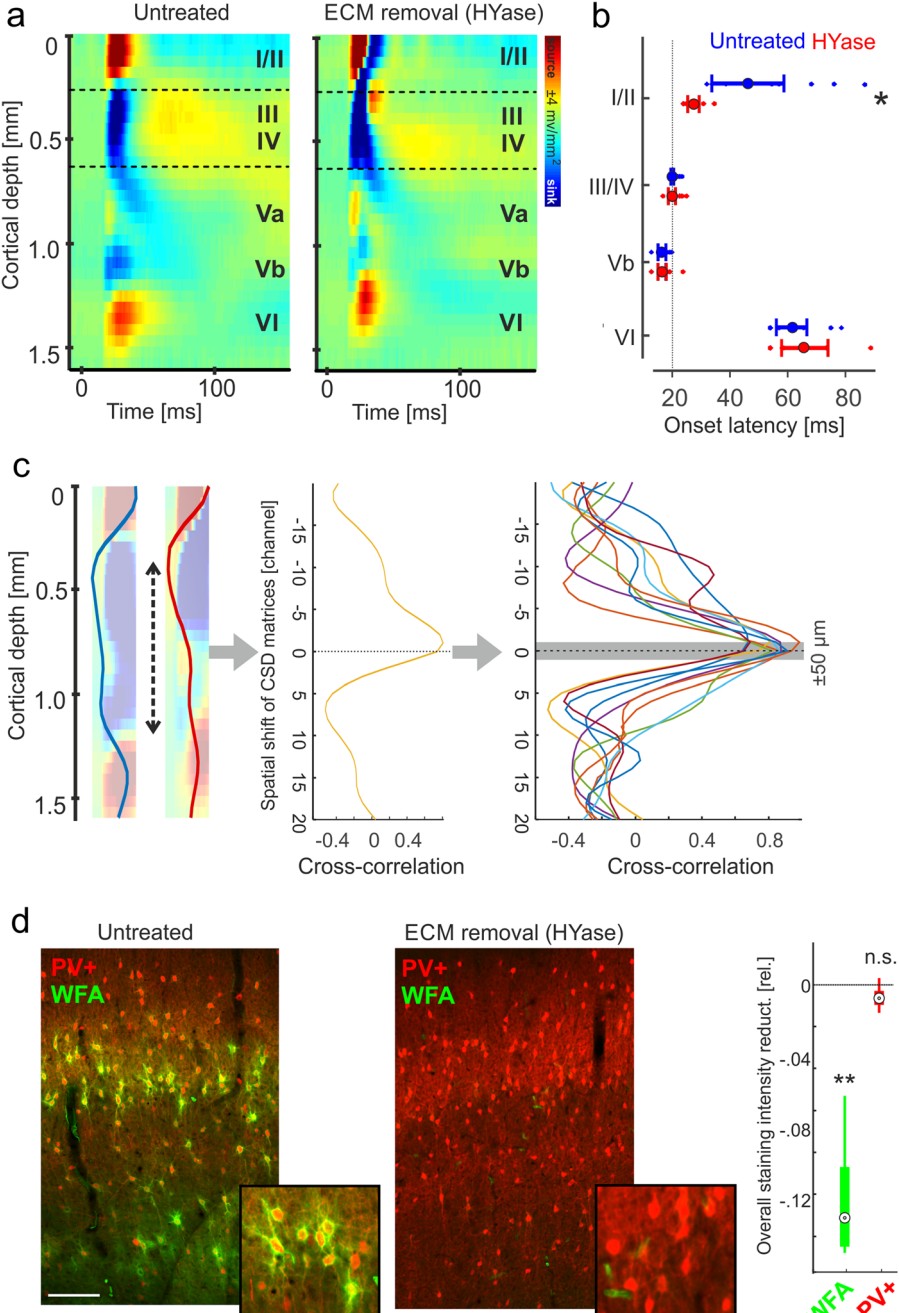

**Fig. 1 ECM weakening changed tone-evoked columnar processing in a layer-dependent manner. a** Tone-evoked CSD profiles display in untreated cortex a canonical feedforward pattern with afferent early sink activity in granular layer III/IV and infragranular layer Vb and subsequent activation of supragranular and infragranular layers. Roman numbers indicate cortical layers. After ECM removal by microinjections of HYase, tone-evoked CSD profiles showed reduced strength of the early afferent input in layer Vb and stronger activation of supragranular layers I/II. **b** Mean onset latencies ±SEM of dominant initial sink components revealed stable onset latencies of early input sinks in cortical layers III/IV and Vb, which are due to thalamocortically relayed input. Current sinks resulting from subsequent intracortical processing showed variability ($n = 9$; *Student's t-test $p < 0.05$). Single dots represent individual data points of each animal. **c** In order to quantify the stability of the cortical laminae along the derivation axis and thus the comparability of the patterns before and after enzyme administration, we have cross-correlated the early onset CSD profile of each animal before and after HYase injection. Relative changes of overlapping sinks and sources of the electric field may occur, while the general spatial profile should be stable. The highest correlation should then be at a zero-lag shift, while shifts of the electrode relative to cortical layers should be detectable by a shift in the peak of the cross-correlogram. Correlation peaks in our data set were all found with at $0 \pm 1$ channel shift corresponding to a maximal shift of $\pm 0.05$ mm. **d** Immunostainings after HYase treatment also revealed reduced density of *WFA* staining and hence reduced PNNs around PV + interneurons. High-resolution confocal microscopy images (Zeiss LSM 700, Germany) show PV (red) and *WFA* (green) staining in the vicinity of the recording site. Stainings reveal layer-dependent densities of PNN structures in the auditory cortex. White scale bar indicates 200 μm. *Right*, Box plots represent the median (circle) and the interquartile range (25% and 75% percentile) and whiskers represent the full range of data ($n = 4$). Gray values of the *WFA* staining were significantly reduced for the HYase-treated side (Student's t-test; $p < 0.002$), but did not differ for PV + ($p > 0.05$).

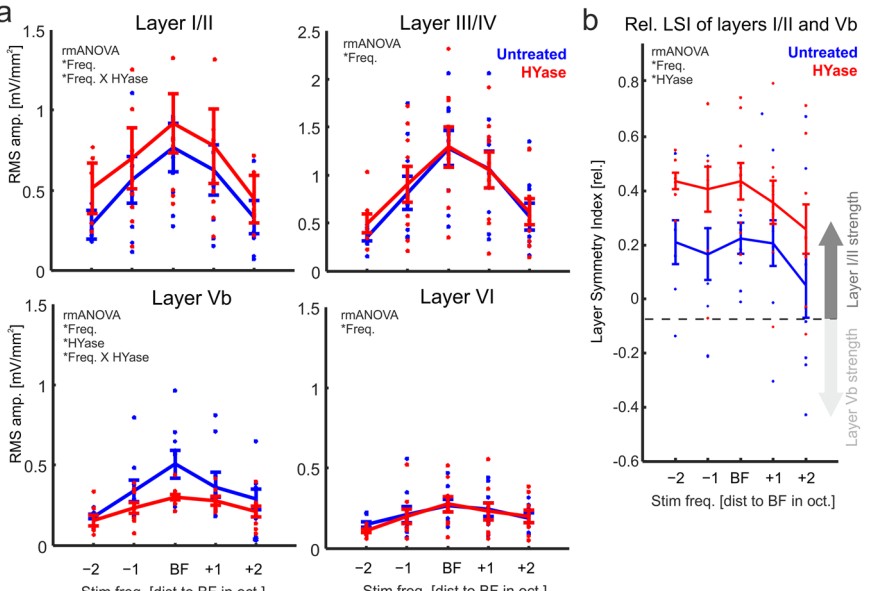

**Fig. 2 Removal of the ECM affects layer-specific processing in the auditory cortex. a** Frequency-response tuning curves for the RMS amplitude (±SEM) of CSD traces from individual cortical layers showed increased activity within supragranular layers I/II and decreased activity in cortical layer Vb (n = 9; represent by individual data points for each animal). Activity in layers III/IV and VI did not change. **b** Correspondingly, the Layer-Symmetry-Index LSI = (I/II – Vb)/(I/II + Vb) showed that after HYase treatment evoked synaptic activity shifted towards supragranular layers independent of the stimulation frequency. Significant main effects and interactions of a 2-way-rmANOVA are indicated in each subpanel (for results see Table 1).

recordings (Fig. 1c). In a previous study, we showed that by local injection at one location the enzyme effectuates a region with a diameter of around 1 mm well covering the recording site[13]. After ECM removal, the feedforward CSD pattern changed in a layer-dependent manner. Tone evoked synaptic responses showed a significant decrease of early infragranular input and a moderate increase of supragranular current flow (Fig. 1a, *right*). No obvious differences were found for layers III/IV and VI. We stained brains after local unilateral HYase injection in temporal cortex against PV (red) and WFA (green) staining (Fig. 1d). High-resolution confocal microscopy (Zeiss LSM 700, Germany) and analysis revealed a layer-dependent accumulation of PNN structures in the ACx particularly around PV + interneurons in cortical layers III/IV of the control injection side, which were significantly reduced at the side of HYase injection. PV + staining density was not significantly altered.

We further quantified tone-evoked root-mean-square (RMS) amplitudes of individual cortical layer activity as a function of stimulation frequency (Fig. 2a). Pure-tone derived frequency response curves showed a consistent decrease of sink activity in infragranular layer Vb induced by HYase injection. The effect was stronger at BF stimulation compared to off-BF stimulation, which indicates a reduced sharpness of spectral tuning. A 2-way rmANOVA correspondingly showed significant main effects for factors tone-frequency ('Freq'), HYase treatment ('HYase'), and a significant interaction 'Freq x HYase' (see Table 1). Thus, besides an overall significant reduction of the Vb response by HYase as main-effect, there was a frequency-specific interaction effect that cannot be explained only by a general modulation of overall excitability. In contrast, the layer I/II RMS-amplitude was moderately, although not significantly increased (p = 0.09; see Table 1). The HYase-induced increase depended on stimulation frequency revealed by a significant interaction effect 'Freq x HYase' in a 2-way rmANOVA (main effect for 'Freq' and a significant interaction 'Freq x HYase'; Table 1). Granular layer

III/IV and infragranular VI activity was unchanged after HYase injection.

To relate the observed changes in layer Vb and I/II to each other we calculated and quantified the relative shift of translaminar columnar activity, we calculated a Layer-Symmetry-Index (LSI) of current flow between cortical layers I/II and Vb (cf. Materials and Methods;). By calculating the LSI = (I/II – Vb)/(I/II + Vb), positive LSI indicate stronger activation of supragranular layers compared to infragranular layers. In untreated cortex, supragranular layers generally display larger evoked synaptic currents compared to infragranular layers with LSI-value around +0.2[26]. HYase treatment, however, amplified this supragranular lead of current flow and LSI values exceeded values of +0.4. Correspondingly, a 2-way rmANOVA of the RMS amplitude-based LSI showed main effects for 'Freq' and 'HYase' without interaction (Table 1). This shows, that the frequency-specific decrease of responses in layer Vb and the moderate, but less specific increase in layer I/II led to a general shift of cortical activity towards upper layers across all stimulation frequencies.

Supragranular sink activity has been related to horizontal, intercolumnar processing[26,31–33]. We have previously developed a method that dissociates intracortically relayed contributions to cortical activity from local circuit activity in order to quantify the cross-columnar activity spread[22,23]. The rationale of this analysis is to analyze the relative residuum of sinks and sources of the extracellular electric field within the local integration cylinder along the linear electrode array radially penetrating the cortical layers. Non-zero values of relative residual CSD (RelResCSD, Eq. 3) contributions, as indicated in Fig. 3a, predominantly results from synaptic input relayed to the cylinder via horizontal intracortical projections. In contrast, the averaged rectified CSD (AVREC, Eq. 2) measures the rectified sum of the sink-source distribution, and hence, reflects the overall cortical activity. We analyzed the frequency-response tuning of the AVREC and the RelResCSD (Fig. 3b). The AVREC showed a moderate, but consistent increase after HYase injection (2-way rmANOVA with

main effects 'Freq' and 'HYase'). The RelResCSD showed a likewise increase that dependent on frequency revealed by a 2-way rmANOVA with main effect 'Freq' and a significant interaction of 'Freq x Treatment' (see Table 1). Both findings are in accordance with increased supragranular activity as a

relative indicator for synaptic input from neighboring cortical columns. Correspondingly, the Q40dB response bandwidth of the AVREC did not change significantly, while the response bandwidth of the RelResCSD was significantly increased (Fig. 3c). Hence, the higher overall activity measured by the AVREC was mainly due to a stronger activation found in supragranular layers I/II leading to an increased corticocortical activity spread measured by the RelResCSD. Comparable changes of the columnar response profile were not found after injection of 0.9% sodium chloride in a set of control animals ($n = 3$; Suppl Fig. 1).

| Table 1 Statistical analysis of layer-wise CSD data by repeated-measures ANOVA. |
| --- |
| **(1) Layer-wise RMS value (Fig. 2a)** |
| RMS Layer I/II |
| Freq.: $F_{4,32} = 4.071$; **p = 0.0156** |
| HYase: $F_{1,8} = 2.014$; p = 0.0936 |
| Freq. X HYase: $F_{4,32} = 3.767$; **p = 0.0231** |
| RMS Layer III/IV |
| Freq.: $F_{4,32} = 6.508$; **p = 0.0055** |
| HYase: $F_{1,8} = 0.257$; p = 0.6267 |
| Freq. X HYase: $F_{4,32} = 7.548$; p = 0.217 |
| RMS Layer Vb |
| Freq.: $F_{4,32} = 4.262$; **p = 0.034** |
| HYase: $F_{1,8} = 7.121$; **p = 0.0283** |
| Freq. X HYase: $F_{4,32} = 4.455$; **p = 0.0283** |
| RMS layer VI |
| Freq.: $F_{4,32} = 3.227$; **p = 0.0478** |
| HYase: $F_{1,8} = 0.641$; p = 0.445 |
| Freq. X HYase: $F_{4,32} = 4.264$; p = 0.169 |
| **(2) Layer-Symmetry (Fig. 2b)** |
| Freq.: $F_{5,4} = 32$; **p = 0.0072** |
| HYase: $F_{3.8,4} = 32$; **p = 0.0264** |
| Frequency X HYase: $F_{5,1} = 8$; p = 0.0571 |
| **(3) AVREC (Fig. 3b)** |
| Freq.: $F_{4,32} = 6.508$; **p = 0.0143** |
| HYase: $F_{1,8} = 6.564$; **p = 0.0335** |
| Freq. X HYase: $F_{4,32} = 3.617$; p = 0.0602 |
| **(4) RelResCSD (Fig. 3b)** |
| Freq.: $F_{4,32} = 18.515$; **p < 0.001** |
| HYase: $F_{1,8} = 11.365$; **p = 0.0098** |
| Freq. X HYase: $F_{4,32} = 6.008$; **p = 0.0100** |
| *All rmANOVAs were Huyn-Feldt corrected. Significant p-values are plotted in bold.* |

**Spectral power of tone-evoked processing changed mainly in infragranular layers**. Next we were interested in the underlying oscillatory nature of the layer-specific synaptic activity. We found that ECM removal had a layer-dependent impact on the spectral power distribution within infragranular layer Vb. Figure 4a shows the grand mean power spectrum (±SEM) of the averaged individual cortical layer activity after BF stimulation (evoked spectrum). To avoid the bias in selecting frequency bands we used the t-values statistical comparison to detect the spectral difference between the experimental conditions. HYase treatment caused significant increase in the evoked spectral responses in infragranular layer Vb in the beta range from 25 to 36 Hz. Spectral power effects might include background effects unrelated to the stimulus presentation. We therefore further calculated the power spectrum during spontaneous recordings. Figure 4b shows the t-values calculated as the mean spectral differences between these conditions across subjects ($n = 9$). The spectrum of spontaneous activity showed no significant difference before and after HYase injection.

**Response dynamics across cortical layers**. We hypothesized that the temporal relationship between activity measured in cortical layers was changed after application of HYase. We further assume that the columnar response was modulated by a differential impact of supragranular and infragranular layers on the laminar response dynamics. To test this, we analyzed this dynamical relationship by employing time-domain Granger causality (GC) estimation on the single trial CSD traces from each cortical layer

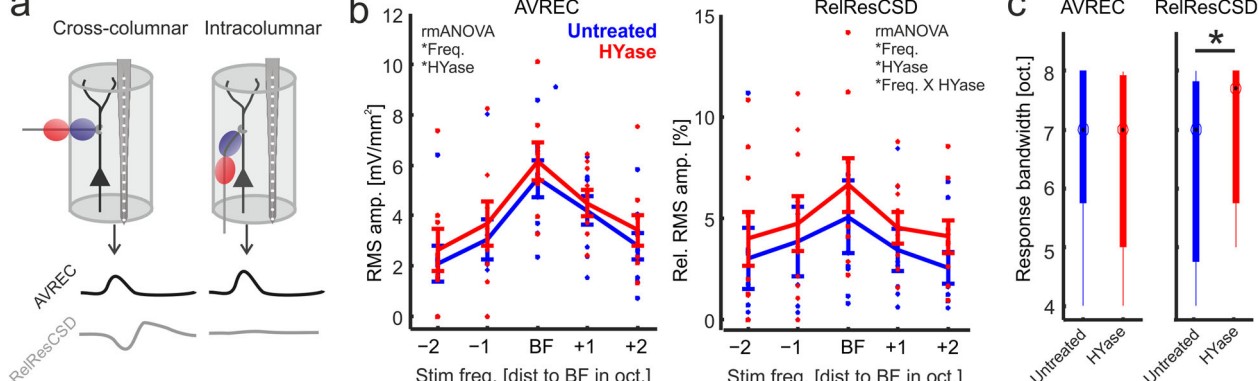

**Fig. 3 Quantification of AVREC and RelResCSD frequency tuning after ECM removal. a** Schema illustrating how the averaged rectified CSD amplitude (black) and the relative residuals of sinks and sources of the extracellular electric field (gray) are reconstructed from linear recording arrays along the depth of the cortical space. Synaptic activity measured within the indicated integration cylinder of the electrode is mainly associated with local columnar activity (AVREC), while the RelResCSD quantifies contributions from lateral corticocortical input. **b** Frequency-response tuning curves of AVREC RMS amplitudes (±SEM) showed increased activity across all stimulation frequencies. RelResCSD showed increase that was depending on stimulation frequency ($n = 9$; represent by individual data points for each animal). Significant main effects and interactions of a 2-way-rmANOVA are indicated in each subpanel by the asterisk * (for results see Table 1). **c** Correspondingly, AVREC showed no change of Q40dB bandwidth (Student's t-test, $p > 0.05$), while RelResCSD bandwidths were significantly increased (*paired Student's t-test, $p = 0.02$). Box plots represent the median (circle) and the interquartile range (25% and 75% percentile). The whiskers represent the full range of data.

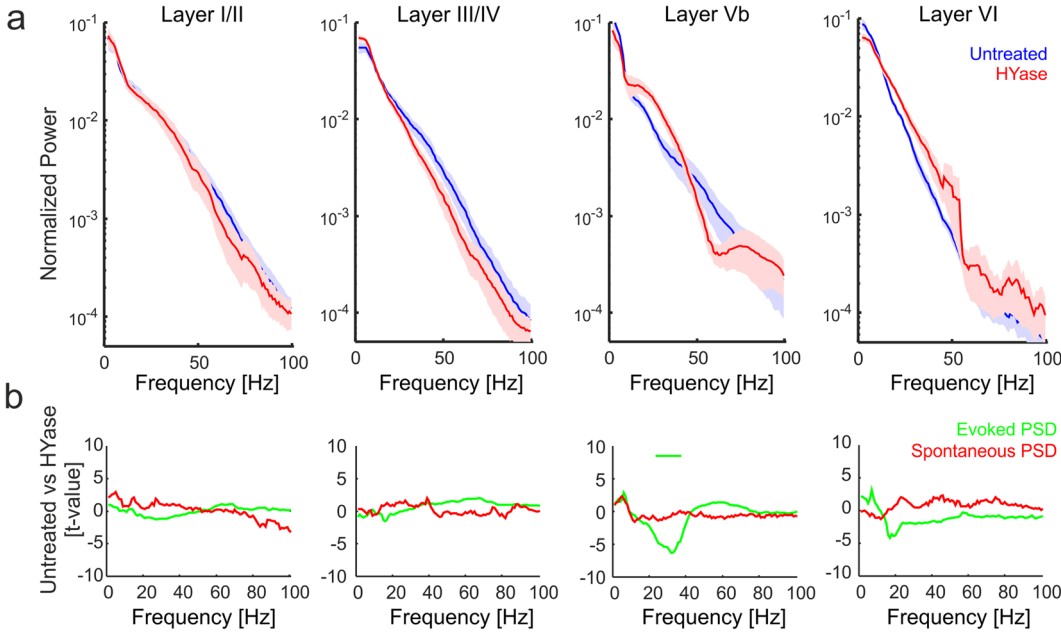

**Fig. 4 Analysis of alterations in layer-dependent spectral power after HYase treatment. a** Layer-dependent analysis of frequency components of evoked responses showed bimodal changes in the spectral power distribution mainly in infragranular layers. In both early and late infragranular layers, beta power showed a significant increase. **b** Standardized differences of evoked (green) and spontaneous (red) power spectra between untreated recordings and after HYase treatment expressed as t-values for each frequency bin (negative t-values indicate higher power after HYase injection). Standardization was based on sample means and standard errors of the mean across subjects ($n = 8$) and tapers ($nt = 5$). Significant differences for the beta range from 25 to 36 Hz for the evoked PSD in infragranular layer Vb are indicated with the green bar and were revealed by a point-by-point Students t-test comparison with a Benjamin Hochberg correction controlling for false discovery rates.

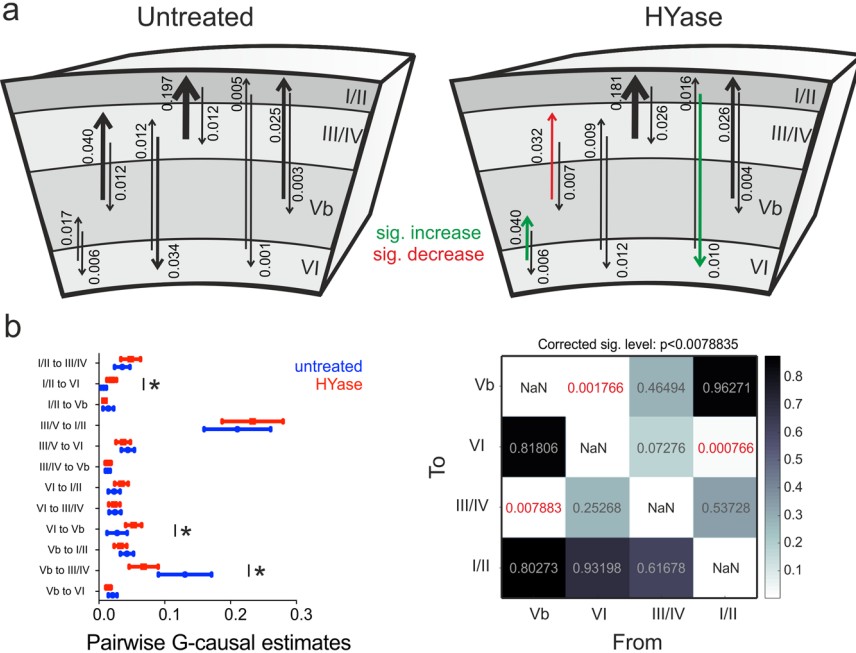

**Fig. 5 Analysis of time-domain pairwise-conditional granger causality.** Granger causality matrices were calculated based on the single trial CSD traces from each cortical layer for each subject ($n = 9$). **a** Median values of the granger causality estimates between cortical layers are shown before (*left*) and after (*right*) HYase treatment. The cortical column scheme depicts the dynamics of the predictive causal relationship across different cortical layers before and after ECM removal. Arrow thickness indicates the median of the causal estimate weights across animals. Red and green arrows indicate where G-causal forecasting undergoes a significant reduction or increase after HYase treatment, respectively. **b** Comparison of the time-domain pairwise-conditional granger causality in the two conditions before and after HYase injection. *Left.* G-causal estimates were plotted with mean and standard error of the mean for each layer pairs from the 9 animals. *Right*, A multiple paired t-test was done on the log transformed values of the estimates and then corrected based on a false discovery rate of 0.05. Significant differences between G-causal measures indicated in the left plot correspond to a corrected level of significance after multiple comparisons of $p < 0.0078835$.

to generate a network model for each animal and then compared the granger causality matrices (Eqs. 4–7). Figure 5a depicts the directed pairwise G-causal estimates between cortical layers by the thickness of connecting arrows and shows the corresponding median values of the G-causal estimate. To compare laminar dynamics before and after application of HYase, we compared the pairwise conditional GC measures. We found a significantly increased predictive power of supragranular activity driving activity within infragranular layer VI (sig. increase indicated by green). Further, predictive power of infragranular layer VI activity for early infragranular layer Vb activity was significantly increased (in green) during evoked columnar responses. This was in contrast to the significant reduction of drive of infragranular layer Vb to the granular layers III/IV (Fig. 5b).

**Layer-dependent changes of spontaneous columnar activity.** Finally, we investigated whether the changes of the columnar responses are specific for sensory evoked activity. We therefore recorded periods of 6 s of spontaneous activity in individual animals ($n = 7$) before and after HYase-induced removal of the ECM. CSD recordings in the untreated auditory cortex revealed the occurrence of spontaneous events of translaminar synaptic information flow, which we refer to as spontaneous columnar events (SCE). SCE's showed a comparable, but distinct synaptic current flow compared to tone-evoked columnar feedforward activation patterns. While tone-evoked responses display earliest synaptic inputs in both granular layers IV and infragranular layer Vb due to afferent thalamocortical input, SCE's showed initial synaptic activity in layer Vb initiating translaminar information flow[34]. Activity in layer Vb was followed by current sink translaminar synaptic activity across the entire column including infragranular and supragranular layers (Fig. 6a, *top*). For quantification of the frequency of SCE's we used a peak detection analysis based on single-trial AVREC traces (see Methods; Fig. 6a, *bottom*). After microinjection of HYase in the vicinity of the recording axis, spontaneous CSD profiles changed considerably. We still observed SCE's, however, with an altered current flow of columnar activity. Individual SCE's were shorter in duration and showed significantly reduced synaptic current flow across all layers (Fig. 6b). ECM weakening also significantly reduced the rate of SCE/s (Fig. 6c). We further attempted to quantify the overall cortical activation based on the overall current flow within the column quantified by the RMS amplitude of the AVREC. We compared this with the RelResCSD quantifying the unbalanced contributions of sinks and sources of the CSD profile, as an indicator for corticocortical spread of activity across columns (see Fig. 3)[22]. While the AVREC showed a consistent significant reduction after HYase injection, the RelResCSD showed a parallel increase indicating relative higher cross-columnar activity spread after treatment (Fig. 6d), which is in accordance with the tone-evoked activity reported before (Fig. 3).

## Discussion

This study investigated the impact of acute removal of the extracellular matrix on the network physiology in the gerbil primary auditory cortex by local injection of the ECM-degrading enzyme hyaluronidase. Based on laminar CSD recordings we found that ECM removal altered the spatiotemporal profile of sensory-evoked synaptic population activity across cortical layers. Stronger activation of supragranular layers I/II was associated with an increased spectral integration via lateral corticocortical activity spread, while tone-evoked activation in infragranular layer Vb was reduced. That further coincided with increased evoked oscillatory power in the beta band (25–36 Hz) within infragranular layer Vb revealed by a multitaper spectral analysis

of layer specific CSD activity. We then applied time-domain Granger causality measures to map the directional causal relationship between cortical layers. We found a significant increase in the GC from supragranular layers I/II towards the infragranular layer VI, and from infragranular layer VI to early infragranular layers Vb. We additionally found a significant decrease in the GC from layer Vb towards granular layers III/IV. Thereby, our data shows that enzymatic removal of the ECM in sensory cortex acutely alters the entire columnar synaptic network processing with stronger weight of supragranular layer activity. Columnar processing is thereby dominated by the integration of cross-columnar widespread activity in upper layers.

**Effects of ECM removal on sensory-evoked activity in auditory cortical circuits.** While we generally found moderately increased activity in supragranular layers across all stimulation frequencies, early synaptic inputs in infragranular layers Vb, related to activity of thalamocortical collaterals, were significantly reduced particularly for BF stimulation (Fig. 2a). Early granular layer III/IV input activity was unaffected implying an asymmetric impact of ECM removal on different thalamocortical input systems in sensory cortex[35]. This is in accordance with reports that sensory input in layer Vb is more vulnerable to juvenile forms of plasticity, as induced by monocular deprivation[36]. Thalamocortical collaterals terminating in infragranular layer Vb also transmit early subcortical inputs (cf Fig. 1) but might be less responsible for the spectral integration across the frequency range at a given cortical patch. Here, granular circuits in auditory cortex have been related to a local amplification of sensory input via recurrent microcircuits yielding a robust bottom-up driven tonotopic tuning[37]. The shift of the entire network activity towards upper layers (see also Fig. 2b) might hence represent an altered and layer-specific input function of spectral information after acute ECM removal. We further found that the overall columnar activity, quantified by the AVREC, was increased, which was also accompanied by a frequency-specific gain increase in the RelResCSD (Fig. 3). We found a corresponding increase of the response bandwidth of the RelResCSD which indicates increased lateral corticocortical activity spread after ECM removal.

With maturation of cortical circuits and phases of experience-dependent shaping of cross-columnar connectivity patterns, intracolumnar circuits stabilize subsequently in order to control its outputs to promote behavior later on[38]. In accordance, learning-related plasticity of synaptic circuits during auditory learning has been linked mainly to corticocortical synaptic connections within supragranular cortical layers I/II[39,40], while thalamic input layers III/IV and Vb show rather moderate plastic changes. Thus, increase of synaptic integration in supragranular layers in the adult cortex after ECM removal may promote plastic adaptations of corticocortical synaptic circuits during learning[26,27,41,42]. We hypothesize that a relative imbalance of synaptic activity between supragranular layers I/II (mediated by corticocortical inputs) and infragranular layer Vb (mediated by early thalamocortical input) is one of the possible physiological mechanisms by which HYase is mediating its action. To further reveal potential network mechanisms of this newly proposed hypothesis, we continued with an analysis of layer-specific neuronal oscillations and translaminar network dynamics.

**Oscillatory layer-specific dynamics after ECM removal.** We found that stimulus-evoked oscillations showed a layer-dependent shift of increased beta power (~25–36 Hz) selectively in infragranular layer Vb (Fig. 4). Infragranular beta oscillations have been linked to altered translaminar processing in line with increased drive from corticocortical inputs in upper layers[43,44]. In

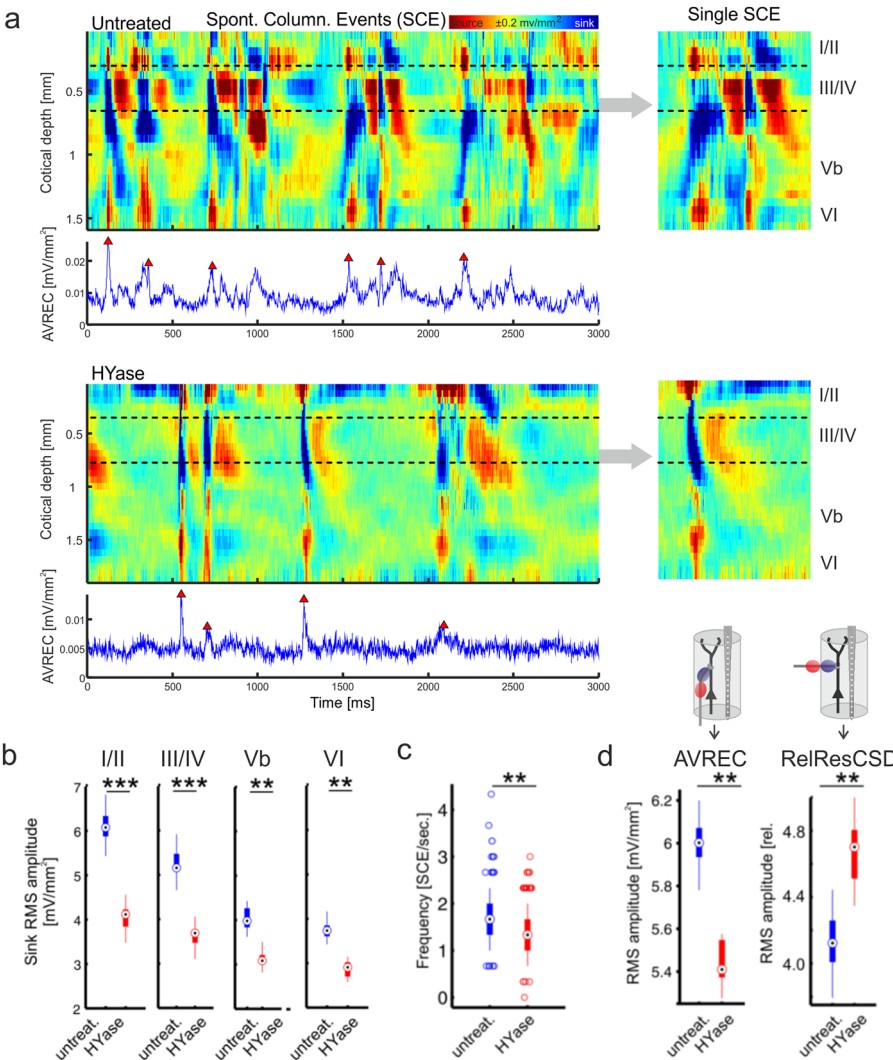

**Fig. 6 Injection of HYase reduced the frequency of spontaneous columnar events and increased cross-columnar activity spread. a** *Top,* Recordings of spontaneous activity showed the occurrence of spontaneous events of translaminar synaptic information flow mimicking evoked columnar feedforward activation profiles called spontaneous columnar events. SCE's in untreated cortex showed information flow starting from pacemaker activity in layer Vb (see inset right). *Bottom,* After Injection of HYase SCE's were shorter in duration, showed no infragranular pacemaker structure and did not expand through all cortical layers (see inlay right). Frequency of SCE's was quantified by peak detection of the AVREC trace based on crossings of 3 SD above baseline (blue traces). **b** Mean RMS amplitudes (±SEM) within separate cortical layers were all significantly reduced after HYase treatment. **c** Frequency of SCE's per second was significantly reduced after HYase treatment. **d** Decreased RMS amplitude of the AVREC ( ± SEM) was paralleled with a significant increase of the RMS amplitude (±SEM) of the RelResCSD. Box plots represent the median (circle) and the interquartile range (25% and 75% percentile). The whiskers represent the full range of data. Plotted single data represent outliers. Asterisks mark significant differences between groups (paired Student's t-test, **$p < 0.01$; ***$p < 0.001$).

the human EEG and animal LFP literature, beta range fluctuations have been related to the deployment of top-down attention[45,46], working memory allocation[47,48], and stimulus salience[49,50]. Increased beta power in deep layers and reverse effects on high-frequency components in upper layers have been specifically linked to altered GABA levels in PV + interneurons in PV-GAD67 mice[51], suggesting that PV cells, generally rich in ECM, might play an important role in balancing neuronal oscillations across cortical layers.

Further, increased layer Vb beta power has been attributed to relatively increased inputs on supragranular distal over infragranular proximal dendrites of intrinsic bursting infragranular neurons[44]. Those neurons therefore might play a pivotal role in orchestrating the translaminar columnar activity. During ECM weakening in the adult cortex, these enhanced low-frequency

oscillations may mimic to a certain degree the orchestration of internal and peripheral inputs present during critical periods of the developing cortex[52–54].

Based on the imbalanced synaptic spatiotemporal pattern and the shorter onset latencies of supragranular activity (Fig. 1b), we hypothesized that supragranular layers exert more drive towards deeper infragranular layers. Time-domain Granger causality analysis further supported our hypothesis of increased drive of supragranular layers I/II towards deeper infragranular layers. Further, we found an altered dynamical relationship of deeper cortical layers. While layer Vb activity became more strongly driven by layer VI input, its output drive onto granular layer III/IV was decreased after ECM removal (Fig. 6). This decrease in feedback drive from infragranular layer Vb to granular layer III/IV is important in controlling the thalamocortical input within granular

layer III/IV[55]. Thereby, ECM removal might induce a higher weight of local sensory input processing and its corticocortical integration and reduced activity of effector circuits towards downstream areas, as motor-related cortical outputs[56–58] potentially supporting higher plastic adaptability towards altered sensory inputs.

**ECM removal as a model for cortical "re-juvenation" and adult learning?** ECM removal has been linked to re-open 'critical periods' of juvenile plasticity unlocking the potential of considerable topographic map adaptations[4,5]. ECM removal, therefore, has been interpreted as a temporally constricted period of neuronal "re-juvenation" promoting a higher level of structural, developmental plasticity[21,59,60]. States of weakened ECM have been also linked to other forms of plasticity, as for instance post-traumatic restoration, recognition memory, seasonal learning in songbirds, and cognitive flexibility during reversal learning[9,10,13,15]. Further, mouse models deficient in specific matrix components, as tenascin-R or brevican, have shown impairments in hippocampus-dependent spatial or contextual acquisition learning[61]. Recently, we showed in wildtype mice that brevican was downregulated 48 h after auditory training in the auditory cortex[14]. Recently, we further showed that such intrinsic reduction of the ECM integrity in the adult cortex can be linked to the stimulation of D1 dopamine receptors suggesting an activity-dependent and learning-related cellular mechanism[16]. Altogether, those studies enunciate the major impact of the integrity of the cortical ECM onto the delicate balance of plasticity and tenacity in the learning brain. Although the underlying physiological circuit mechanisms are still rather elusive, our study suggests that imbalanced supragranular and infragranular synaptic activity during states of weakened ECM may play a major role in promoting cross-columnar integration, which has been related to learning-related plasticity in several systems[27,41,42,62–65].

In this study, we investigated the effects of ECM removal acutely and under ketamine-xylazine anesthesia. We have recently shown that, in addition to generally very similar spatiotemporal activity patterns, anesthesia mainly influences gain enhancement in layer III/IV at best frequency stimulation[29]. Particularly, ketamine-xylazine had only minor effects on the spectral beta range and off-BF-evoked activity, which showed strongest effects after HYase treatment. In this study, ECM removal specifically enhanced beta power in infragranular layer Vb, and minor effects on layer III/IV activity.

Beta power is also higher in children than adults[66], which we hence found to be reversed after "re-juvenation" by HYase (Fig. 5). Furthermore, the reduction in spontaneous columnar events during states of weakened ECM might mimic decreased states of spontaneous synchronicity in juvenile cortex[67,68], where peripheral stimulation dominates temporal discharge patterns[69]. Spontaneous columnar activity in ACx in anaesthetized animals has been linked to a dominant drive of pyramidal neurons and interneurons in deeper layers[34]. We suppose that ECM removal altered the balance of activity between these cell types reflected in fewer spontaneous translaminar events, but higher cross-columnar spread compared to the untreated adult cortex[65]. The function of earliest activity in sensory cortex is to establish topographic maps based on experience-dependent plasticity of sensory bottom-up input via thalamocortical circuits[53,70,71]. Early sensory-derived synaptic plasticity is hence pronounced in cortical layers II–IV, but less affecting infragranular synapses[68,72,73]. It has been argued before that long-range corticocortical processing in upper layers of the juvenile visual cortex is necessary to correlate activity across cortical space in order to promote such early bottom-up-driven sensory-based plasticity[33,70,74]. Consistently, two consecutive days of strabismus

cause selective strengthening of horizontal connections in layers II/III between cortical columns representing the same eye[75]. The acute cortical state after ECM weakening in our study may resemble such a processing mode by the increase of relative residual contributions of the spontaneous CSD patterns indicating increase of corticocortical activity spread. Further research in awake, behaving animals would be needed to link such circuit activity after removal of the ECM in sensory cortex directly to the unlocking of plastic adaptations of layer-specific network dynamics in the adult brain (cf. Fig. 6). Of note, behavioral effects of ECM modulations are most prominent after a certain time of training[10,13]. Hence, they may have no immediate impact on perception and behavior. Nevertheless, we have unraveled a potential circuit mechanism, that potentially allows the cortical network to adjust the balance of internal activity and sensory-driven inputs[76]. Ultimately, such changes may promote developmental[5] and learning-derived plasticity underlying increased performance in cognitively demanding tasks and memory recall over time[3,60,77]. Understanding the biology of such circuit readjustments due to translaminar synaptic weight adaptations might also help to better understand the role of the adult ECM during pathological conditions as sensory loss or developmental neurological disorders[68,78].

## Methods

**Experimental Design.** Experiments were performed on 12 adult male ketamine-xylazine anesthetized Mongolian gerbils (age: 4–10 months, body weight: 80–100 g; $n = 9$ HYase injection, $n = 3$ control injections). All experiments were conducted in accordance with the international NIH Guidelines for Animals in Research and with ethical standards for the care and use of animals in research defined by the German Law for the protection of experimental animals. Experiments were approved by an ethics committee of the state Saxony-Anhalt, Germany.

**Surgery and electrophysiological recordings.** Surgical and experimental procedures have been described before[27,29]. In detail, Mongolian gerbils ($n = 12$) were anesthetized by an initial dose of 50 mg/KG ketamine and 5 mg/KG xylazine intraperitoneal and kept under narcosis via an infusion (0.06 ml/h) of 45% ketamine (50 mg/ml, Ratiopharm, Germany), 5% xylazine (Rompun, 2%, BayerVital, Germany) and 50% isotonic sodium chloride solution (154 mmol/l, Braun, Germany). Status of anesthesia was monitored, and body temperature was kept at 37 °C. The right ACx was exposed by craniotomy (~3 × 4 mm) of the temporal bone. Recordings were performed in an acoustically and electrically shielded recording chamber. Laminar profiles of local field potentials (LFP) were measured using linear 32-channel-shaft electrodes (NeuroNexus, 50 μm inter-channel spacing, 413 μm² site area; type A1 × 32–5 mm-50-413) inserted perpendicular to the cortical surface. Neuronal potentials were preamplified (500x), band-pass filtered between 0.7 and 170 Hz (3 dB cut-off frequency), digitized at 2 kHz (Multichannel Acquisition Processor, Plexon Inc.) and averaged over 40–80 stimulus repetitions. The location of the field AI in primary ACx was identified by vasculature landmarks and physiological parameters[22]. In this study, we analyzed only data from recordings which exceed 90 min after implantation of the electrode and the glass capillary (see next section)–a state where the cortical responses under anesthesia have been largely stabilized as reported recently[29].

**Enzymatic removal of the extracellular matrix by intracortical injections.** In direct proximity of the recording electrode, we inserted a thin glass capillary (<250 μm distance) approximately within the middle of the right ACx in depths of 500 μm in order to perform microinjections (Nanoliter injector 2000, WPI) of HYase at the cortical patch of recording in ACx. We started with recording of baseline activity 90 min after insertion of micropipette and electrode. All relaxation movements of the brain relative to the recording electrode and glass pipette should be completed in order to guarantee stabilized cortical responses[29]. Then, we recorded spontaneous and tone-evoked activity over a period of roughly 1 h. After that, we injected 500 nl of HYase-solution (500 units) in 22 steps (22.8 nl each) with 3 s pause in between[13]. Before we continued recordings, we waited at least 2.5 h until we continued with recordings after ECM removal.

**Current source density and residue analysis.** One-dimensional current-source density (CSD) profiles were calculated from the second spatial derivative of the LFP:[25,79]

$$\sim CSD \approx \frac{\delta^2\theta(z)}{\delta z^2} = \frac{\theta(z+n\Delta z) - 2\theta(z) + \theta(z-n\Delta z)}{(n\Delta z)^2} \tag{1}$$

where $\theta$ is the field potential, z the spatial coordinate perpendicular to the cortical laminae, $\Delta z$ the spatial sampling interval (50 µm), and $n$ the differentiation grid. LFP profiles were smoothed with a weighted average (Hamming window) of 7 channels (corresponding to a spatial filter kernel of 300 µm; linear extrapolation of 4 channels at boundaries; see Happel et al.[22]). CSD profiles reveal the patterns of extracellular current influx (sinks) and efflux (sources) of positive charges. Positive charge influx results from the depolarization of postsynaptic neurons by synaptic populations in laminar neuronal structures, while sources to a large degree reflect passive return currents[25]. CSD activity thereby reveals the spatiotemporal sequence of neural activation across cortical layers as ensembles of synaptic population activity. Due to the extracellular negativity, current sinks are depicted in blue and current sources in red.

Main sink components were found to represent the architecture of primary sensory input from the lemniscal auditory part of the thalamus–the medial geniculate body (MGB). Lemniscal thalamocortical projections from the ventral division of the MGB terminate on pyramidal neurons with local dendritic arbors and ramifying axons in cortical layers III/IV, which are henceforth referred to as the granular input layers. Collaterals of these thalamocortical projections also target infragranular layer Vb[28,35,80]. Layers above (I/II) or beyond (V–VI) the granular layers are generally referred to as supragranular and infragranular layers, respectively[81]. In our study, corresponding sink activity within cortical layers were referred to as granular activity (III/IV) supragranular activity (layer I/II), and infragranular activity (layer Vb and VI). Root-mean square (RMS) amplitudes of individual layers were determined for individual channels and then averaged.

Based on single trial CSD profiles without spatial filtering we transformed the CSD by rectifying and averaging waveforms of each channel (n) comprising the laminar CSD profile (AVREC). The AVREC waveform provides a measure of the temporal pattern of the overall strength of transmembrane current flow[79]. The relative residue of the CSD (RelResCSD), defined as the sum of the non-rectified divided by the rectified magnitudes for each channel, was used to quantify the balance of the transmembrane charge transfer along the recording axis:[22]

$$AVREC = \frac{\sum_{i=1}^{n}|CSD_i|(t)}{n} \qquad (2)$$

$$RelResCSD = \frac{\sum_{i=1}^{n}CSD_i(t)}{\sum_{i=1}^{n}|CSD_i|(t)} \qquad (3)$$

We have shown before that the RelResCSD provides a quantitative measure of the contributions of intercolumnar, corticocortical synaptic circuits to the overall measured local cortical activity. In short, the rationale is that corticocortical circuits would be more likely distributed beyond the integration cylinder surrounding the electrode array in which extracellular currents contribute to the measured LFP (cf. Fig. 3). We quantified the RMS values of both parameters during the period of stimulus presentation (200 ms).

**Auditory stimulation and estimation of tuning curves**. We presented pseudo-randomized series of pure tones (200 ms with 5 ms sinusoidal rising and falling ramps; spanning 8 octaves from 250 Hz to 32 kHz; 40 dB over threshold; inter-stimulus interval: 600 ms; digitally synthesized using Matlab and converted to analog signals by a data acquisition National Instruments card; PCI-BNC2110). Stimuli were delivered via a programmable attenuator (g.PAH, Guger Technologies; Austria), a wide-range audio amplifier (Thomas Tech Amp75) and a loud-speaker (Tannoy arena satellite) positioned in 1 m distance to the animal's head. The response threshold was determined as the lowest intensity eliciting a significant response at any frequency 2 SD over baseline (>5 ms). The sound level then was adjusted to 40 dB above threshold, which generally was at intensities around 65 dB SPL. Response bandwidths of AVREC and RelResCSD were quantified as the Q40dB-values corresponding to the number of octaves evoking a tone-evoked response over the threshold criterion[23]. The best frequency (BF) of the recording site was determined as the frequency evoking the highest RMS amplitude within the granular layer III/IV at 40dB above threshold. For estimation of pure-tone based frequency response curves, evoked RMS amplitudes of individual parameters were binned into frequency bins with respect to their distance to the best frequency of each individual recording (−3–4 oct.; −1–2 oct.; BF; + 1–2 oct.; +3–4 oct.). Cross-laminar activation patterns were quantified and statistically analyzed by introducing a Layer-Symmetry-Index (LSI). The LSI quantifies the total activity bias based on RMS values of the supragranular layer vs. infragranular layer Vb by calculating LSI = (I/II − Vb) / (I/II + Vb). A positive LSI indicates that columnar synaptic activity is biased towards stronger supragranular activity, while negative values indicate stronger recruitment of infragranular synaptic circuits. For pure tone processing, no hemispheric differences between left and right auditory cortex are expected for either evoked response properties, nor behavioral differences[22,82,83].

**Recording and analysis of spontaneous activity**. We recorded spontaneous data in 7 animals for at least 3 minutes before and after injection of HYase. Spontaneous CSD data was cut into 6 s traces and we analyzed RMS values of the AVREC, RelResCSD and layer dependent CSD traces of individual sink components (see above). Spontaneous events of translaminar synaptic information were referred to as spontaneous columnar events (SCE). The frequency of SCE's was quantified by a

peak detection of the AVREC trace based on crossings of 3 standard deviations (SD) above baseline (median value of the whole spontaneous trace) and at least 150 ms between individual peaks.

**Spectral analysis of CSD data**. Traditional Fourier spectral analysis uses a single windowing function, which yields large variance and biases especially at higher frequencies. We therefore used the multitaper Fourier transform employing multiple splenial tapers that are orthogonal to each other (Chronux toolbox; http://chronux.org), as explained in detail elsewhere[24]. CSD-signals were transformed into the frequency domain by a multitaper FFT (600 ms epochs) and 5 tapers of 600 ms length with a time-bandwidth product of 3 and no padding were employed[84]. The stimulus evoked power spectrum was averaged by the complex valued spectra across trials before calculating the power as squared magnitude of the average. To normalize the power spectral values, we divided it by the sum of all values. As induced spectra might include background effects unrelated to the stimulus, we further calculated the power spectrum during spontaneous recordings. Finally, the mean and the standard error of the mean of the power spectra were calculated across subjects. Power spectral comparison before and after HYase administration was done across subjects, separately for the BF condition and layer (I/II, III/IV, Vb, VI). Comparisons were restricted to a frequency range from 1 to 100 Hz. To avoid biased determination of power spectrum bands, t-scores were calculated between recordings in untreated cortex and after HYase injection for each spectrum, frequency, and layer using sample mean and variance and the significance level was corrected using a Benjamin Hochberg procedure as a suitable method to adjust the false discovery rate when performing a large number of multiple comparisons[85]. Accordingly, we ranked individual p-values in an ascending order. Critical Benjamin-Hochberg (BH) p-values are calculated by $p^* = (i_r/m)*Q$, where $i_r$ is the individual p-value's rank, m is the total number of tests, and Q is the false discovery rate (0.1). The critical BH $p^*$ value serves as adapted level of significance.

**Time-domain conditional granger causality**. To identify the dynamical nature of the directed connectivity between different cortical layers as defined by the CSD analysis, we calculated the multivariate conditional causality in the time-domain using MVGC Matlab toolbox[86]. Generally, unconditional Granger causality (GC) assumes two time series (X, Y) and provides information about the magnitude of their causal dependency. Briefly, the unconditional GC measures if past values of the time series Y provide information that predicts upcoming values X better than the past history of X itself. Mathematically, this is done by constructing and comparing two linear vector autoregressive models (VAR). the first model described as a full model where the value of X at particular time point is a linear weighted sum of both its own and Y's past values according to the following equation:

$$X(t) = \sum_{k=1}^{p} A_{xx,k}.X_{t-k} + \sum_{k=1}^{p} A_{xy,k}.Y_{t-k} + \varepsilon_{x,t} \qquad (4)$$

where p is the model order, $A_{xx,k}$ and $A_{xy,k}$ are the full regression coefficient matrices and $\varepsilon_{x,t}$ is the error or residual process. The second model is described as the reduced model where we only use X past values to predict X current values.

$$X(t) = \sum_{k=1}^{p} A'_{xx,k}.X_{t-k} + \varepsilon'_{x,t} \qquad (5)$$

By taking the logarithmic ratio between the covariance of the two residuals we can get the estimate F describing Y being G-causal of X.

$$F_{Y \to X} = ln\frac{Cov(\varepsilon'_{x,t})}{Cov(\varepsilon_{x,t})} \qquad (6)$$

In other words, G-causal quantifies the reduction in the prediction error when including Y past values in the model. The main limitation of the unconditional GC is the spurious causal relationship that can be derived if exist a third variable Z that both X and Y depends on. the conditional GC takes into account the inter-dependent nature of the jointly distributed multivariate times series data by including the variables Z in the reduced and full regression model and now calculating the conditional G-causal F between X, Y is explained as the following

$$F_{Y \to X|Z} = ln\frac{Cov(\varepsilon'_{x,t})}{Cov(\varepsilon_{x,t})} \qquad (7)$$

The multivariate granger causality toolbox adds to the standard G-causal estimation as it allows for reduced model parameter computation from the full model parameters for more computationally effective and accurate statistical estimates[86]. We used the toolbox to estimate the conditional granger causality from the single trial evoked CSD dataset. To ensure that our data was stationary, we took the first time-derivative of CSD traces and then tested for stationarity using the Augmented Dickey Fuller test. After that, we estimated the order of our vector autoregressive (VAR) model using Akiko information criteria (AIC). The model order is then used to estimate the model parameters (the regression coefficients and residual errors) using the "LWR algorithm" or the multivariate extensions to Durbin recursion. The autocovariance sequence derived from the full model parameters is estimated and used to derive the reduced VAR parameters using an algorithm to solve the Yule-Walker equation. The residual covariance matrices of both the reduced and full model are used to estimate the G-causal as in Eq. 7. The

G-causal estimates for each animal is then log transformed, and a paired t-test is performed. A false discovery rate of 0.05 was set for multiple comparison and a critical p-value is determined as a cut off for significance[85].

**Immunohistochemistry.** To assess effects of HYase injection on the ECM, we applied microinjections in naive mature gerbils (≥3 month; $n = 4$) under anesthesia as explained above (5 mg ketamine and 3 mg xylazine per 100 g body weight, i.p.). We injected HYase in the ACx unilaterally and used 0.9% saline at the opposite side as a control. Animals were transcardially perfused with 20 ml phosphate buffered saline (PBS) followed by 200 ml paraformaldehyde (4% in PBS) 2.5 h after injections in order to assess the ECM removal at the earliest time point where recordings have been done in the experimental animals. Brains were removed, postfixated overnight, cryoprotected in 30% sucrose solution for 48 h, frozen, and cut in a series of horizontal sections (40 μm thickness) on a cryostat (Leica CM 1950; Germany). After blocking (5% goat serum) and washing, the sections were stained using *Wisteria floribunda* agglutinin coupled to fluorescein (WFA; 1:100; cat# FL-1351, RRID: AB_2336875, Vector Laboratories, CA, USA) for 12 h (at 4 °C). Then, sections were incubated with anti-PV (mouse, 1:4000, 12 h, cat# 235, RRID: AB_10000343, Swant, Switzland) and afterwards with a Cy3-conjugated anti-mouse secondary antibody (goat, 1:500, 2 h, cat#115–165–166, RRID: AB_2491007, Dianova, Germany). Finally, sections were mounted on gelatin-coated slides and coverslipped with MOWIOL (Fluka, Germany). Images were taken using a confocal microscope (Zeiss LSM 700, Germany) equipped with a 2.5x objective (NA 0.085, Zeiss, Germany).

For the analysis of WFA and PV staining intensities, microscopic images of regions of interest were taken with normalized exposure time and laser power for each animal as described above. RGB image channels were splitted and converted non-weighted into 8-bit gray-scale images; then, the general background was substracted and the images inverted (ImageJ, v. 1.52i, NIH, Bethesda, MD, USA). Mean gray values of the staining intensities in the ACx (all layers) of the HYase-treated and the contralateral side injected with 0.9% sodium-chloride were measured in 6 sections of 200 μm interval of 4 animals using the Measurement Tool of ImageJ.

**Statistical analysis and reproducibility.** Comparison of multiple groups was performed by multifactorial repeated-measures ANOVAs (rmANOVAs). For comparison between two groups, paired sample Student's t-tests were used. Generally, a significance level of α = 0.05 was chosen. For multiple comparisons, we used Bonferroni-corrected levels of significance.

**Reporting summary.** Further information on research design is available in the Nature Research Reporting Summary linked to this article.

## Data availability

Preprocessed data are available via the following GitHub repository: https://github.com/CortXplorer/Eltabbal_et-al. Raw data (plx-files and converted Matlab-files) are available upon request from the corresponding authors.

## Code availability

All code is available via the following GitHub repository: https://github.com/CortXplorer/Eltabbal_et-al.

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

## Acknowledgements

The work was supported by grants from the Deutsche Forschungsgemeinschaft DFG (SFB779, FR 2758/3–1, HA 6753/3–1) and the Leibniz-Society WGL granted to RF and MFKH.

## Author contributions

M.T. and M.F.K.H. conceived the study and designed the experiments. Experiments were carried out by M.T. and supervised by M.F.K.H. M.T., M.D., and M.F.K.H. analyzed data. M.D. developed statistical test procedures. J.H. and E.B. performed histological experiments. H.N. and R.F. supplied agents. Figures were prepared by M.T. and M.F.K.H. M.T., M.D., and M.F.K.H. wrote the paper with the assistance from all co-authors.

## Funding

## Competing interests

The authors declare no competing interests.
