## [Peer Review File · Communications Biology]

Reviewers' comments:

Reviewer #1 (Remarks to the Author):

In the manuscript, Happel MFK. and colleagues studied the effect of extracellular matrix (ECM) removal on the synaptic network processing in adult auditory cortex (ACx). ECM in the ACx of Mongolian gerbils was acutely digested by local microinjection of hyaluronidase (HYase) and laminar recordings of local field potentials and current-source density (CSD) was used to quantify the spatiotemporal sequence of spontaneous and stimulus-evoked layer-specific synaptic activity. Authors found that ECM removal altered the spatiotemporal profile of sensory-evoked synaptic activity across cortical layers. Authors propose that there was a stronger activation of supragranular layers I/II and reduced tone-evoked activation in infragranular layer Vb. In addition, there was increased evoked oscillatory power in beta oscillations (25-36 Hz) within the infragranular layer Vb. The findings of the effects of ECM modulation on the sensory integration and translaminar cortical network dynamics are also novel and might be useful in developing therapeutic approaches targeting ECM. However, several concerns should be addressed related to the presentation and interpretation of the results. Few minor grammatical errors are also noted.

Major concerns:

1. In Fig 1 it is not clear what are red, yellow vs blue colors? Does blue indicates higher or lower activity? What is the scale?
2. In Fig 2B increase in LSI is primarily driven by decreased Vb responses not increased LI/II responses as LI/II amplitude is not significantly different between Control and HYase condition (Fig 2A). This should be stated more clearly.
3. The statement "In contrast, layer I/II RMS-amplitude was increased in a frequency-specific manner" is confusing as similar to Vb the effects were stronger at BF stimulation compared to off-BF (Freq), but the effects of HYase was not significant ($p=0.09$), suggesting that similar to layers III/IV, layer I/II activity was also unchanged after HYase injection. Authors state again "This (LSI-value increase) shows, that the frequency specific decrease of responses in layer Vb and increase in layer I/II were counterbalanced", but the increase in I/II amplitude was not statistically significant and the increase in LSI-value is primarily driven by a reduction in Vb.
4. In Fig.3, right panel N was not listed for t-test. What are the circles and triangles depict on the graph?
5. In Fig 4 legend authors state "Asterisks mark the significant differences for the beta range from 25-36 Hz for the evoked PSD in infragranular layer Vb" but no asterisks are shown in the figure.
6. In Fig 5A increases for Vb to VI and I/II to Vb comparisons should be shown in different color (green for example) than decrease. Arrows are also hard to see and should be larger in size.
7. In Fig.5C, right panel p value of $p<0.005994$ for I/II to Vb comparison is greater than 0.003558, which is indicated as significant difference value in the legend, but is shown as significant in the graph (red color). Different color scheme for this panel would also prevent a confusion in comparison to panel B. In addition, it should be stated what shades of green are representing in panels B and C.
8. The statement "we found a significant increase in the GC from supragranular layers I/II towards the infragranular layer Vb" was based on G-causal estimates for each layer pairs (Fig5C, left panel), not a multiple paired t-test (Fig5C, right panel), which shows $p<0.005994$ for I/II to Vb comparison, which is greater than significant value $p<0.003558$.

Minor

1. A punctuation is missing in Introduction section before "In a seminal study..."
2. Full name should be given in the text for the abbreviated terms mentioned for the first time such as BF, AVREC, RelResCSD, 2SD (Results section, and if Materials and Methods section goes after Results section).

3. In the Results section, "Layer-dependent changes of tone-evoked synaptic activity", Fig.1 A or B panel should be specified.
4. In the Materials and Methods all equations should be numbered, so it would be easier to find an appropriate equation, which authors refer to in the text.
5. What does "frontal sections" mean in Immunohistochemistry section of Materials and Methods? Are these coronal sections?
6. Define what "nt" is in Results section.
7. In Literature cited, titles of the papers should be presented consistently, all words capitalized for all citations or not capitalized. There is a mixture of both there.
8. Fig. 2A: Axis Y scale for Layers I/II, Vb and VI should be the same to appreciate the layer-specific differences.

Reviewer #2 (Remarks to the Author):

In this manuscript, the authors use electrophysiological recordings in the auditory cortex to determine the effects of enzymatic removal of the extracellular matrix on the spatio-temporal dynamics of cortical columnar activity evoked by sound. Overall, the analyses show that degradation of the ECM does change the response patterns in the auditory cortex, with effects that are interpreted to indicate greater cortico-cortical activation (compared to thalamocortical/intracolumnar activity). Interestingly, the effects of ECM removal extended also to spontaneous events (i.e., not sound-driven) that likewise show greater cross-columnar activity (though fewer spontaneous events).

Overall, the findings are interesting and the experiments are well put together and well controlled. The authors should also be commended for frequency-specific analyses, which are informative. There are a few points that the authors should address:

1. The title does not seem to describe the core finding of increased cross-columnar activity; it does so only indirectly. It may be better to revise the title to more directly address the main finding.
2. Is it possible to obtain "after" ECM degradation recordings, e.g., after the enzyme is "washed out" or otherwise metabolized so to see a return of "normal" connectivity and activity?
- 3A. It would be helpful to explain the acronyms and conceptual utility of AVREC and RelResCSD within this manuscript. It is difficult to understand the significance of differences (or failures to find differences) in these two measures of activity without going back to several papers to even learn what the acronyms mean. The figure (Fig. 3, left side panel) is there, but not entirely helpful. These measures need to be justified and explained since they are at the core of many of the analyses.
- 3B. The authors may wish to include their recent publication, which is obviously relevant to the current manuscript. (Zempeltzi et al., 2020 Commun Biol).
4. Were HYase injections in the right or the left hemisphere? If in both (in different subjects), is there a hemispheric effect?
5. It is not always clear how many N are included in the groups. For example, in the figures, how many recordings are in each plot? As a directed example in Figure 2 or in Figure S1., it may be better to plot individual data points on the line graphs instead of standard error bars. That will give a better representation of the raw data and easily indicate "N".

6. Figure 2B y-axis right-side label of "...leading" is confusing. Do the authors mean "strength", i.e., of the cumulative response?

Minor:

- There are some typos and difficult sentence constructions that should be corrected, e.g., last paragraph of the Discussion "Most importantly, behavioral effects of ECM.....which let us suppose, that they may have no immediate effect...." or page 2 of Results, last paragraph, "RelResCSD should a likewise increase that DEPENDED on" (instead of "DEPENDENT")

- Figure 2B y-axis is mislabeled as "Layer Similarity Index" instead of "Layer Symmetry Index"

- Figure 5 arrows showing directionality of the relationships are extremely difficult to see and should be enlarged.

Reviewer #3 (Remarks to the Author):

El-Tabbal and colleagues investigate an interesting question: how does the removal of the extracellular matrix in primary auditory cortex of gerbils influence layer specific response dynamics. Albeit the motivation for this question is not entirely clear based on the abstract or introduction, it is still an interesting scientific endeavor. However, after looking at the laminar CSD figures (1 and 6, and supplemental figure 1), it became apparent that the main findings are profoundly influenced by a relative shifting of the brain relative of the recording electrode. Therefore, even given the relatively sophisticated signal processing methods, the results are hard to interpret for the reasons detailed below.

1) Pre- and post-injection recordings are not comparable based on the laminar CSD figures (1 and 6, and supplemental figure 1) and the description of methods. The insertion of the glass capillary and/or the injection of HYase-solution renders pre- and post-injection laminar results incomparable. This could be either due to a "contraction" of cortical layers due to injection, or simply settling of the recording electrode after insertion of the glass capillary. For example, the layer 6 infragranular sink almost moves to layer 5b in Fig 1 in untreated compared to treated animals. This alone would result in apparent layer specific response amplitude changes. A similar, approximately .3-.5 mm shift is observable in figures 6 and supplemental figure 1 (control injection).

2) Related to the above, how were cortical layers determined?

3) What is the evidence for any oscillatory activity in the beta band? What is the significance of "stimulus-evoked oscillatory power decrease" in the beta band in infragranular layer Vb? It can likely be explained by the apparent recording electrode shift (major concern 1). Showing "raw" ERP (CSD) response for each of the layers would be helpful in judging this.

4) If the effect of extracellular matrix removal is frequency specific, what indicates the use of time-domain Granger causality analysis (as opposed to frequency domain GC)? Also, an increased amplitude ERP response in the supragranular layers is likely to bias GC measures.

5) What does "weakened ECM" mean exactly (first paragraph of results)? In general, the manuscript would benefit from more precisely formulating the results and supporting them with quantitative analyses.

Response Letter for COMMSBIO-20-1891

Manuscript: The extracellular matrix regulates cortical layer dynamics and cross-columnar frequency integration in the auditory cortex

Response to the reviewers

We would like to thank the reviewers for their careful reading and constructive inputs. We believe that this has led to a significantly improved manuscript. Please find attached to this submission the revised manuscript that addresses all concerns of the reviewers.

The revised version now includes new data and experiments, updated figures and new panels, new supplemental information, and a substantially rewritten manuscript. We also adapted the referencing according to the format of the journal.

All changes we are referring to in the response letter are referenced by page and paragraph of the current version to facilitate their identification in the revised manuscript. In the provided reviewer's version of the revised manuscript, all major changes are highlighted in **yellow** for your and the reviewers' inspection. We also hand in a clear copy of the manuscript.

In summary, the revision led to the following new figures in the manuscript:

NEW Fig. 1 – new analysis of electrode track stability and histological quantification (in response to Reviewer#3)

NEW Fig. 2 – plotting single data points within bar graphs (in response to Reviewer#2)

NEW Fig. 3 – plotting single data points within bar graphs and correction of panel c (in response to Reviewer#1)

NEW Fig.5 – new depiction of the Granger causality measure (in response to Reviewers#1 and 2)

NEW Supplementary Fig. 1 – includes electrode stability measure for control groups (in response to Reviewer#3)

Please find below the point-by-point responses (in blue) to each comment of the reviewers (in black).

Response to Reviewer 1

The manuscript, Happel MFK. and colleagues studied the effect of extracellular matrix (ECM) removal on the synaptic network processing in adult auditory cortex (ACx). ECM in the ACx of Mongolian gerbils was acutely digested by local microinjection of hyaluronidase (HYase) and laminar recordings of local field potentials and current-source density (CSD) was used to quantify the spatiotemporal sequence of spontaneous and stimulus-evoked layer-specific synaptic activity. Authors found that ECM removal altered the spatiotemporal profile of sensory-evoked synaptic activity across cortical layers. Authors propose that there was a stronger activation of supragranular layers I/II and reduced

tone-evoked activation in infragranular layer Vb. In addition, there was increased evoked oscillatory power in beta oscillations (25-36 Hz) within the infragranular layer Vb. The findings of the effects of ECM modulation on the sensory integration and translaminar cortical network dynamics are also novel and might be useful in developing therapeutic approaches targeting ECM. However, several concerns should be addressed related to the presentation and interpretation of the results. Few minor grammatical errors are also noted.

We would like to thank the reviewer for the careful reading and the appreciation of our study. Please find changes according to the helpful comments and suggestions raised by the reviewer in our revised manuscript.

Major concerns:

1. In Fig 1 it is not clear what are red, yellow vs blue colors? Does blue indicates higher or lower activity? What is the scale?

We agree that this was not sufficiently explained. We now make this more explicit on page 9, 1st column:

“CSD profiles reveal the patterns of extracellular current influx (sinks) and efflux (sources) of positive charges. Positive charge influx results from the depolarization of postsynaptic neurons by synaptic populations in laminar neuronal structures, while sources to a large degree reflect passive return currents²⁵. CSD activity thereby reveals the spatiotemporal sequence of neural activation across cortical layers as ensembles of synaptic population activity. Due to the extracellular negativity, current sinks are depicted in blue and current sources in red.”

We similarly state this in the first part of the results (page 2, 2nd column, 2nd par.) to make it comprehensible for the reader to understand Figure 1.

2. In Fig 2B increase in LSI is primarily driven by decreased Vb responses not increased I/II responses as I/II amplitude is not significantly different between Control and HYase condition (Fig 2A). This should be stated more clearly.

We now more precisely state in that paragraph that the effect in layer Vb was significant and therefore we resume on page 3, 2nd column:

“In contrast, the layer I/II RMS-amplitude was moderately, although not significantly increased ($p=0.09$; see Tab. 1). The HYase-induced increase depended on stimulation frequency revealed by a significant interaction effect ‘Freq x HYase’ in a 2-way rmANOVA (main effect for ‘Freq’ and a significant interaction ‘Freq x HYase’; Tab. 1.1). Granular layer III/IV and infragranular VI activity was unchanged after HYase injection.”

3. The statement “In contrast, layer I/II RMS-amplitude was increased in a frequency-specific manner” is confusing as similar to Vb the effects were stronger at BF stimulation compared to off-BF (Freq), but the effects of HYase was not significant ($p=0.09$), suggesting that similar to layers III/IV, layer I/II activity was also unchanged after HYase injection. Authors state again “This (LSI-value increase) shows, that the frequency specific decrease of responses in layer Vb and increase in layer

I/II were counterbalanced”, but the increase in I/II amplitude was not statistically significant and the increase in LSI-value is primarily driven by a reduction in Vb.

In light with the reply to point 2, we agree that this statement also needs revision. The term ‘counterbalanced’ was inappropriate. Indeed, we believe that the increased shift towards upper layer activity, revealed by the change of the Layer-Symmetry Index, did not depend on frequency (evident from the main effects for ‘Frequency’ and ‘HYase’, but without significant interaction). This happened despite the fact, that both single layer RMS measures – from layers Vb and I/II – showed such interaction effects, which led to our initial formulation. We now more specifically say, that the effects in the cortical layers I/II and Vb depended on the stimulation frequency, but that the LSI measure in sum revealed a general shift of cortical activity towards upper layer across all stimulation frequencies.

This section in the revised manuscript now reads (page 3, 2nd column ff.):

“Correspondingly, a 2-way rmANOVA of the RMS amplitude-based LSI showed main effects for ‘Freq’ and ‘HYase’ without interaction (Tab. 1). This shows, that the frequency-specific decrease of responses in layer Vb and the moderate, but less specific increase in layer I/II led to a general shift of cortical activity towards upper layers across all stimulation frequencies.”

4. In Fig.3, right panel N was not listed for t-test. What are the circles and triangles depict on the graph?

We thank the reviewer for carefully reading the manuscript. We now state in the caption of Figure 3 that box plots represent the median (circle) and the box indicates the interquartile range (25% and 75% percentile). The whiskers represent the full range of data. The statistical results are now shown in the caption and also referred to in the main text. The triangles were plotted erroneously and are removed in the revised version.

5. In Fig 4 legend authors state “Asterisks mark the significant differences for the beta range from 25-36 Hz for the evoked PSD in infragranular layer Vb” but no asterisks are shown in the figure.

The caption was wrong. Significant differences are indicated by the green bar covering the spectral range of significant difference within the beta range from 25-36 Hz. We have corrected the caption in the revised version.

6. In Fig 5A increases for Vb to VI and I/II to Vb comparisons should be shown in different color (green for example) than decrease. Arrows are also hard to see and should be larger in size.

We would like to thank the reviewer for this suggestion. We have considerably revised the Granger causality analysis. Please note that we found an erroneous allocation of infragranular layers Vb and VI in our previous analysis. In the revised analysis we find, consistent with our hypothesis, a reduced drive from early infragranular input layers Vb towards the granular input layers III/IV, while supragranular layers increase their causal drive onto infragranular layers VI. We also changed the representation of the data substantially (see below in this response) and hope that it now has gained clarity. We incorporated panels **a** and **b** to better visualize the direction in a cortical pictogram. Please also see corresponding point 8 of the reviewer.

7. In Fig.5C, right panel p value of $p < 0.005994$ for I/II to Vb comparison is greater than 0.003558, which is indicated as significant difference value in the legend, but is shown as significant in the graph (red color). Different color scheme for this panel would also prevent a confusion in comparison to panel B. In addition, it should be stated what shades of green are representing in panels B and C.

We changed the figure panel c (now b) accordingly.

8. The statement “we found a significant increase in the GC from supragranular layers I/II towards the infragranular layer Vb” was based on G-causal estimates for each layer pairs (Fig5C, left panel), not a multiple paired t-test (Fig5C, right panel), which shows $p < 0.005994$ for I/II to Vb comparison, which is greater than significant value $p < 0.003558$.

We corrected the level of significance and changed the figure and the caption accordingly. We also show the new figure 5 including caption here for the reviewer’s direct inspection:

Fig. 5. Analysis of time-domain pairwise-conditional granger causality. Granger causality matrices were calculated based on the single trial CSD traces from each cortical layer for each subject ($n=9$). **a** Median value of the granger causality estimates between cortical layers are shown before (left) and after (right) HYase treatment. The cortical column scheme depicts the dynamics of the predictive causal relationship across different cortical layers before and after ECM removal. Arrow thickness indicates the median of the causal estimate weights across animals. Red and green arrows indicate where G-causal forecasting undergoes a significant reduction or increase after HYase treatment, respectively. **b**. Comparison of the time-domain pairwise-conditional granger causality in the two conditions before and after HYase injection. Left. G-causal estimates were plotted with mean and standard error of the mean for each layer pairs from the 9 animals. Right, A multiple paired t-test was done on the log transformed values of the estimates and then corrected based on a false discovery rate of 0.05. Significant differences between G-causal measures indicated in the left plot correspond to a corrected level of significance after multiple comparisons of $p < 0.0078835$.

Minor

1. A punctuation is missing in Introduction section before “In a seminal study...”

Done.

2. Full name should be given in the text for the abbreviated terms mentioned for the first time such as BF, AVREC, RelResCSD, 2SD (Results section, and if Materials and Methods section goes after Results section).

We carefully checked to introduce each term at first use.

3. In the Results section, “Layer-dependent changes of tone-evoked synaptic activity”, Fig.1 A or B panel should be specified.

We significantly revised Figure 1 and now specified each figure panel in the text.

4. In the Materials and Methods all equations should be numbered, so it would be easier to find an appropriate equation, which authors refer to in the text.

We made the according changes and referenced equations in the text.

5. What does “frontal sections” mean in Immunohistochemistry section of Materials and Methods? Are these coronal sections?

We indeed made a mistake here. The figure shows a horizontal section, which is now corrected.

6. Define what “nt” is in Results section.

In the previous version, this indicated the ‘number of tapers’, which was not necessary at this point. We now omitted to introduce it as abbreviation and only write it in the methods section.

7. In Literature cited, titles of the papers should be presented consistently, all words capitalized for all citations or not capitalized. There is a mixture of both there.

We thank the reviewer for this remark and have revised the reference list accordingly.

8. Fig. 2A: Axis Y scale for Layers I/II, Vb and VI should be the same to appreciate the layer-specific differences.

We see the point by the reviewer and thank for this comment. We have prepared a revised version of the figure accordingly. Also, according to the Figure Guidelines of Communications Biology, we have now prepared all figures that present data with mean±SEM including single data points to also indicate the n.

Response to Reviewer 2

In this manuscript, the authors use electrophysiological recordings in the auditory cortex to determine the effects of enzymatic removal of the extracellular matrix on the spatio-temporal dynamics of cortical columnar activity evoked by sound. Overall, the analyses show that degradation of the ECM does change the response patterns in the auditory cortex, with effects that are interpreted to indicate greater cortico-cortical activation (compared to thalamocortical/intracolumnar activity). Interestingly, the effects of ECM removal extended also to spontaneous events (i.e., not sound-driven) that likewise show greater cross-columnar activity (though fewer spontaneous events).

Overall, the findings are interesting and the experiments are well put together and well controlled. The authors should also be commended for frequency-specific analyses, which are informative. There are a few points that the authors should address:

We would like to thank the reviewer for carefully reading and appreciating our study. Please find changes according to the constructive comments and suggestions raised by the reviewer in our revised manuscript.

1. The title does not seem to describe the core finding of increased cross-columnar activity; it does so only indirectly. It may be better to revise the title to more directly address the main finding.

Based on the reviewer's input, we have rephrased the title, which now reads: "The extracellular matrix regulates cortical layer dynamics and cross-columnar frequency integration in the auditory cortex".

2. Is it possible to obtain "after" ECM degradation recordings, e.g., after the enzyme is "washed out" or otherwise metabolized so to see a return of "normal" connectivity and activity?

Here, the reviewer raises an important point. In this study, it was our goal to describe changes of acute ECM degradation on the microcircuit level. We believe, that we do not induce changes of the overall gross anatomical connectivity patterns during this acute state of reduced ECM, but rather a functional change of relative translaminar connectivity. We have shown before, that it requires, for instance, substantial daily behavioral training in order to provoke learning-dependent changes after days (Happel et al. 2014). However, reduction of the ECM has been shown to enhance spontaneous changes of spine morphology acutely around 1h after treatment (de Vivo et al., 2013). This has been linked to shifted synaptic activity properties, which are the most likely source of the translaminar changes in the recruitment of synaptic activity patterns *in vivo*, as reported in our study. We measured this effect roughly 2.5h after enzyme injection. Based on previous *in vitro* results (de Vivo et al., 2013) and our own, now revised and improved, histological analysis (see revised Fig. 1), we assume, that the enzymatic process of ECM degradation at this time was already on an advanced level with significant ECM reduction. In parallel, the enzyme is presumably already inactivated to a significant degree after 1h (Bikbaev et al., 2015; Fig S4). In that sense, we already reported on a 'wash-out'-condition in this experimental setting, as we analyzed a state, where the enzymatic ECM degradation was significantly present, while the enzyme was presumably not active anymore to a substantial degree. This state of degraded ECM now would be present for up to 10 days to 2 weeks, until the ECM would gradually reinstate (Happel et al., 2014b). In order to show, that after ECM regrowth the initial cortical columnar physiology would also readjust, one would need to record from

the same location 2-4 weeks later with chronically implanted electrodes, which bears methodological concerns (glial growth at insertion site) and goes beyond the scope of this study.

3A. It would be helpful to explain the acronyms and conceptual utility of AVREC and RelResCSD within this manuscript. It is difficult to understand the significance of differences (or failures to find differences) in these two measures of activity without going back to several papers to even learn what the acronyms mean. The figure (Fig. 3, left side panel) is there, but not entirely helpful. These measures need to be justified and explained since they are at the core of many of the analyses.

We now included a more comprehensive rationale of the residual current-source density analysis. In previous studies, we developed this method to dissociate the thalamocortically relayed from the intracortically relayed contributions to cortical activity (Happel et al., 2010; Happel and Ohl, 2017). The method is based on the analysis of the residuum of the sinks and sources of the extracellular electric field reconstructed from measurements of the local field potential along linear electrode arrays radially penetrating the cortical layers. Using pharmacological blocking of intracortical transsynaptic relayed activity, we have demonstrated that a non-vanishing residuum of the CSD, i.e. a non-zero net current in a Gaussian cylinder surrounding the electrode array axis; as indicated in Figure 3a), predominantly results from extracellular currents relayed to the cylinder via lateral intracortical projections. The method is highly sensitive and allows, for example, determining the tuning with stimulus frequency of the relative contributions of thalamocortically and intracortically relayed activity to a tonotopic site in the auditory cortex (cf. Happel et al., 2010).

We have now added this more detailed explanation to the revised manuscript on page 4, 1st column, which now reads:

“Supragranular sink activity has been related to horizontal, intercolumnar processing^{26,31–33}. We have previously developed a method that dissociates intracortically relayed contributions to cortical activity from local circuit activity in order to quantify the cross-columnar activity spread^{22,23}. The rationale of this analysis is to analyze the relative residuum of sinks and sources of the extracellular electric field within the local integration cylinder of along the linear electrode array radially penetrating the cortical layers. Non-zero values of relative residual CSD (RelResCSD, equation 3) contributions, as indicated in Fig. 3a, predominantly results from synaptic input relayed to the cylinder via horizontal intracortical projections. In contrast, the averaged rectified CSD (AVREC, equation 2) measures the rectified sum of the sink-source distribution, and hence, reflects the overall cortical activity.”

Furthermore, we have also increased the clarity of the caption of Figure 3a.

3B. The authors may wish to include their recent publication, which is obviously relevant to the current manuscript. (Zempeltzi et al., 2020 Commun Biol).

We appreciate the reviewer’s comment and have cited our recent publication on page 7, 1st-2nd column:

“Although the underlying physiological circuit mechanisms are still rather elusive, our study suggests that imbalanced supragranular and infragranular synaptic activity during states of weakened ECM may play a major role for promoting cross-columnar integration, which has been related to learning-related plasticity in several systems^{27,41,42,62–65}”

4. Were HYase injections in the right or the left hemisphere? If in both (in different subjects), is there a hemispheric effect?

In this study we investigated the right auditory cortex. For pure tone stimulation, there are no hemispheric differences in the gerbil based on the evoked physiology, nor behavioral learning parameters (Ohl and Scheich, 1997; Ohl et al., 1999; Happel et al., 2010; Jeschke et al., 2020)(e.g. Ohl et al., 1997; Happel et al, 2010; Jeschke et al., 2020). We have now explained this on page 10, 1st column. We now state this explicitly in the methods section and also included a statement in the revised manuscript on page 10, 1st column:

"For pure tone processing, no hemispheric differences between left and right auditory cortex are expected for either evoked response properties, nor behavioral differences^{22,82,83}."

5. It is not always clear how many N are included in the groups. For example, in the figures, how many recordings are in each plot? As a directed example in Figure 2 or in Figure S1., it may be better to plot individual data points on the line graphs instead of standard error bars. That will give a better representation of the raw data and easily indicate "N".

We thank the reviewer for their careful reading. We have now added the information of the *n* to each caption and added single data points to all plots where this was applicable.

6. Figure 2B y-axis right-side label of "...leading" is confusing. Do the authors mean "strength", i.e., of the cumulative response?

We have made the according change.

Minor:

- There are some typos and difficult sentence constructions that should be corrected, e.g., last paragraph of the Discussion "Most importantly, behavioral effects of ECM.....which let us suppose, that they may have no immediate effect...." or page 2 of Results, last paragraph, "RelResCSD should a likewise increase that DEPENDED on" (instead of "DEPENDENT")

We substantially edited the manuscript and had a native speaker proof-reading the final version. We corrected typos and checked to introduce all technical abbreviations at the first place, where they occur. Sentences longer than 30 words were aimed to be divided into two.

- Figure 2B y-axis is mislabeled as "Layer Similarity Index" instead of "Layer Symmetry Index"

We have made the according change.

- Figure 5 arrows showing directionality of the relationships are extremely difficult to see and should be enlarged.

We have considerably revised the Granger causality analysis and now display it illustrated by a cortical layer scheme. Please note that we found an erroneous allocation of infragranular layers Vb and VI in our previous analysis. In the revised analysis we find, consistent with our hypothesis, a

reduced drive from early infragranular input layers Vb towards the granular input layers III/IV, while supragranular layers increase their causal drive onto infragranular layers VI. Please find the revised Figure 5 also attached to this response for the reviewer's direct inspection:

Fig. 5. Analysis of time-domain pairwise-conditional granger causality. Granger causality matrices were calculated based on the single trial CSD traces from each cortical layer for each subject ($n=9$). **a** Median values of the granger causality estimates between cortical layers are shown before (*left*) and after (*right*) HYase treatment. The cortical column scheme depicts the dynamics of the predictive causal relationship across different cortical layers before and after ECM removal. Arrow thickness indicates the median of the causal estimate weights across animals. Red and green arrows indicate where G-causal forecasting undergoes a significant reduction or increase after HYase treatment, respectively. **b**. Comparison of the time-domain pairwise-conditional granger causality in the two conditions before and after HYase injection. *Left*. G-causal estimates were plotted with mean and standard error of the mean for each layer pairs from the 9 animals. *Right*, A multiple paired t-test was done on the log transformed values of the estimates and then corrected based on a false discovery rate of 0.05. Significant differences between G-causal measures indicated in the left plot correspond to a corrected level of significance after multiple comparisons of $p < 0.0078835$.

Response to Reviewer 3

In El-Tabbal and colleagues investigate an interesting question: how does the removal of the extracellular matrix in primary auditory cortex of gerbils influence layer specific response dynamics. Albeit the motivation for this question is not entirely clear based on the abstract or introduction, it is still an interesting scientific endeavor. However, after looking at the laminar CSD figures (1 and 6, and supplemental figure 1), it became apparent that the main findings are profoundly influenced by a relative shifting of the brain relative of the recording electrode. Therefore, even given the relatively

sophisticated signal processing methods, the results are hard to interpret for the reasons detailed below.

We would like to thank the reviewer for the careful reading and the appreciation of our study. Please find changes according to the helpful comments and suggestions raised by the reviewer in our revised manuscript.

1) Pre- and post-injection recordings are not comparable based on the laminar CSD figures (1 and 6, and supplemental figure 1) and the description of methods. The insertion of the glass capillary and/or the injection of HYase-solution renders pre- and post-injection laminar results incomparable. This could be either due to a “contraction” of cortical layers due to injection, or simply settling of the recording electrode after insertion of the glass capillary. For example, the layer 6 infragranular sink almost moves to layer 5b in Fig 1 in untreated compared to treated animals. This alone would result in apparent layer specific response amplitude changes. A similar, approximately .3-.5 mm shift is observable in figures 6 and supplemental figure 1 (control injection).

The reviewer's questions reveal weaknesses in the methodological description of the initial manuscript and are hence justified. We now explain more explicitly the experimental approach. Further, we have performed another set of data analyses in order to proof recording track comparability between conditions. We included this new data analysis into the revised manuscript and commented on these important questions.

First, we have improved the description of the experimental procedure. At the beginning of each experiment, we implanted the recording electrode and in close proximity the glass capillary in the auditory cortex. We waited >90min before we started measuring the baseline condition. After this time, all relaxation movements of the brain relative to the recording electrode after the implantation should be completed to our experience (Happel et al., 2014a; Brunk et al., 2019; Deane et al., 2020). Therefore, we can exclude movements of the electrode relative to the brain due to tissue relaxation in the measurements before and after enzyme injection. See this explanation also within the revised manuscript on page 8, 2nd column.

The changes in the sink-source patterns relative to the recording channels the reviewer is referring to, are rather explained by an altered spatiotemporal superposition of current sinks and sources of the electrical field before and after enzyme administration than by a shift of the recording track. Accordingly, the differences of the CSD patterns reflect the physiological changes in spatiotemporal synaptic responses revealed by our study.

In order to quantify the stability of the cortical laminae along the derivation axis and thus the comparability of the patterns before and after enzyme administration, we have developed a statistical correlation method. This method is based on the assumption, that, although relative changes and resulting superpositions of sinks and sources may occur, the general profile of sinks and sources across recording channels should be rather stable. Particularly, the thalamocortical input activity after best-frequency stimulation resulting in the initial tone-evoked current sinks are the most reliable neuronal observables in order to link recording channels to the corresponding thalamocortical-recipient layers III/IV and Vb (cf. Happel et al., 2010). In a first attempt, we have analyzed the onset latencies of the corresponding sink components. In case of a considerable shift of the recording track, we would assume to find clearly observable shifts of the onset latencies at the same recording channels, as they would shift in relation to the cortical layers receiving short-latent thalamocortical input. As shown in a new set of data analysis, we observed such onset latency shifts of sink components only in later, subsequent sink components that are due to synaptic activity routed intracortically in layers I/II and VI. Onset latencies of initial thalamocortically relayed sink

components in layers III/IV and Vb are highly stable across both conditions, which rules out considerable shifts of the recording track along the time line of the experiment.

Furthermore, we have used the entire 32-channel CSD profile and calculated a channel-wise average over the first 50 ms after tone onset, which includes the short-latent tone-evoked response. We then cross-correlated the resulting matrices from before and after ECM removal along the derivation axis with a shift in space. If we would assume a shift of the recording track, we should expect a peak of the correlogram at the exact shift of the recording track (for instance, with an inter-channel distance of 50 μ m, a 200 μ m shift would correspond to a maximal correlation with a 4-channel shift). However, as the highest correlation was found at zero shift of the matrices, we can assume comparability of the spatial distribution of sinks and sources at the initial tone-evoked component before and after the injection. Importantly, this method is compensating for relative changes of the response amplitudes, but take into account the general spatial profile of sinks and sources.

This new analysis clearly shows stability of the recording tracks (peak of the correlation at 0 ± 1 channel shift; corresponding to a shift of $\pm 50 \mu$ m) within all of our individual recordings. Also, in the control animals with injection of sodium-chloride, we have shown in Supplemental Figure 1, the estimated shift was not larger (see new panel Suppl. Fig. 1b).

We believe that particularly due to the substantial impact of the HYase on the spatiotemporal distribution of current sinks and sources, this new analysis is of general relevance for the study. We also included it as a main figure into a revised version of Figure 1 in the manuscript and a corresponding description on page 3, 1st column. For the reviewer's inspection, we include the Figure 1a-c and the corresponding caption in this response.

New panel Fig. 1. ECM weakening changed tone-evoked columnar processing in a layer-dependent manner. a Tone-evoked CSD profiles display in untreated cortex a canonical feedforward pattern with afferent early sink activity in granular layer III/IV and infragranular layer Vb and subsequent activation of supragranular and infragranular layers. Roman numbers indicate cortical layers. After

ECM removal by microinjections of HYase, tone-evoked CSD profiles showed reduced strength of the early afferent input in layer Vb and stronger activation of supragranular layers I/II. **b** Mean onset latencies of dominant initial sink components revealed stable onset latencies of early input sinks in cortical layers III/IV and Vb, which are due to thalamocortically relayed input. Current sinks resulting from subsequent intracortical processing showed variability (n=9; *Student's t-test p<0.05). Single dots represent individual data points of each animal. **c** In order to quantify the stability of the cortical laminae along the derivation axis and thus the comparability of the patterns before and after enzyme administration, we have cross-correlated the early onset CSD profile of each animal before and after HYase injection. Relative changes of overlapping sinks and sources of the electric field may occur, while the general spatial profile should be stable. Highest correlation should then be at a zero lag shift, while shifts of the electrode relative to cortical layers should be detectable by a shift in the peak of the cross-correlogram. Correlation peaks in our data set were all found with at 0 ± 1 channel shift corresponding to a maximal shift of ± 0.05 mm.

2) Related to the above, how were cortical layers determined?

Our rationale is based on laminar LFP recordings and CSD analysis, which we have used in previous reports to identify lemniscal thalamocortical input layers based on the early tone-evoked sink components. In Happel et al. (2010) we have shown for instance that early synaptic inputs persist after intracortical silencing with the GABA_A-agonist muscimol. They could further be related histologically to the cortical layers III/IV and Vb and also IVa receiving direct thalamocortical inputs from the MGB (Deliano et al., 2018; Brunk et al., 2019; Deane et al., 2020) This is in further accordance with a detailed analysis of the cortical CSD pattern in ACx reported by others (Schaefer et al., 2015). In the current study, we did not distinguish VIa and VIb and subdivided deeper cortical layers into Vb and VI. We adapted the manuscript to make this more clear to the reader. Our revised text on page 2, 2nd column ff. now reads:

"Recordings showed a canonical feedforward processing pattern with main short-latent current sink components in granular layers III/IV and infragranular layer Vb most prominently for stimulation with the best frequency (BF; Fig. 1a, left). Those short-latent sinks reflect the synaptic afferent input from the lemniscal auditory part of the thalamus, the ventral medial geniculate body (vMGB), with main projections to granular layers and collaterals within layer Vb²⁶⁻²⁸. This has been revealed by earlier studies showing persistent thalamocortical inputs in corresponding layers after intracortical silencing with the GABA_A-agonist muscimol^{22,23,29} in accordance with reports by others^{28,30}. Early sinks were followed by synaptic activation of supragranular (I/II) and deep infragranular (VI) layers (Fig. 1a), which are due to intracortical synaptic processes²²."

3) What is the evidence for any oscillatory activity in the beta band? What is the significance of "stimulus-evoked oscillatory power decrease" in the beta band in infragranular layer Vb? It can likely be explained by the apparent recording electrode shift (major concern 1). Showing "raw" ERP (CSD) response for each of the layers would be helpful in judging this.

The reviewer here raises an important point concerning the functional nature of the observed effects.

Fig. 4 shows the power spectral densities of the evoked responses in different layers, i.e. the CSD-response time- and phase-locked to the stimulus, before and after application of HYase. After HYase injection power significantly increased in the high beta range between 25 and 36 Hz. This power increase cannot be explained by an electrode shift. Firstly, the aforementioned cross-correlation analysis of CSD profiles before and after HYase injection provides evidence of stable electrode

positions. Furthermore, electrode shifts would have resulted necessarily in spectral changes across all of the cortical layers under investigation. To the contrary, effects, were highly layer-specific and only during evoked activity.

We furthermore found our effects to be treatment-specific. Before the injection of HYase, spectral power distributions reflect a typical 1/F distribution (power law). Any delineation from this linear decay, as described for layer Vb after HYase injection, is indicative of a peak structure at a specific oscillatory component. The statistical testing then revealed the band-limited effect of this change in oscillation within the beta range 25-36 Hz. For the shown example in Fig. 1 of our study, we have also attached the raw event-related CSD-curves from each of the cortical layers before and after HYase treatment in this response for the reviewer's inspection (see below Fig. R1). These curves reveal the oscillatory nature of the CSD-responses.

In fact, our interpretation of the data in the previous version of the manuscript was not sufficiently clear. Distinct infragranular beta power has been related to a certain integration pattern of synaptic inputs along the proximal and distal dendrites spanning the deeper and upper layers, respectively (Sherman et al., 2016). In this work, the authors specifically show that strong and temporally congruent input to the distal, upper dendritic parts over the deeper proximal part of infragranular pyramidal neurons increases their specific contribution to transient beta events. Indeed, we find a temporally more convergent appearance of upper and deeper layer inputs due to shorter onset latencies of synaptic activity in the upper cortical layers I/II (see mean onset latency of individual sink components; new Fig. 1b). While deeper layer inputs were significantly reduced after ECM removal, the shifted Layer-Symmetry-Index was indicative for a stronger lead of upper layer circuits to the tone-evoked columnar response profile. While early inputs in the deeper layers are reminiscent of thalamocortical sensory input, supragranular layer activity can be attributed to corticocortical integration processes (Happel et al., 2010) potentially with respect to top-down information (Brunk et al., 2019).

We summarize that ECM removal led to increased cross-columnar inputs in upper layers, which presumably shifted the integrated distal and proximal inputs on infragranular layer pyramidal neurons towards upper layers leading to a layer-specific beta power increase, in agreement with Sherman et al., (2016). This circuit description now underlines our interpretation that the ECM regulates the columnar integration of more distributed intracortical input and local sensory input processing via altered translaminar cortical network dynamics.

Figure R1. CSD profiles before and after HYase removal (left) and corresponding raw CSD traces from each of the cortical layers under investigation (right).

4) If the effect of extracellular matrix removal is frequency specific, what indicates the use of time-domain Granger causality analysis (as opposed to frequency domain GC)? Also, an increased amplitude ERP response in the supragranular layers is likely to bias GC measures.

We agree that it would be interesting to assess the frequency-specificity of the causal interactions and their change with ECM degradation. However, a prerequisite of Granger causality analysis is to transform the non-stationarities in the data to a near-stationary one. To correct for the non-stationarities we took the first derivative in time domain, which was empirically proven to be the best method to correct for non-stationarities in our data. Though, differencing causes a change in the spectral properties of the signal. Hence frequency-domain GC interpretation might be much more challenging.

The main aim of our time-domain granger causality analysis is to understand the dynamics of causal relations between different layers before and after the degradation of the ECM. The earlier onset latency shift and stronger synaptic weights in supragranular layers suggested that we will find a dynamical changes in the relationship between layers with supragranular layers taking more stronger lead. And this can be revealed in the time-domain. We have considerably revised the Granger causality analysis. Please note that we found an erroneous allocation of infragranular layers Vb and VI in our previous analysis. In the revised analysis we find, consistent with our hypothesis, supragranular layers increase their causal drive onto infragranular layers VI, while a reduced drive from early infragranular input layers Vb towards the granular input layers III/IV.

In order to rule out that a mere (yet not significant) increase in ERP amplitude in supragranular layers may trivially change or bias the GC measures, we used simulated CSD data. We run the GC estimation on the same data of the untreated measurements and a version, where we upscaled the supragranular CSD traces by a factor of 5. When comparing this with the original untreated data, we found no significant difference in the causal estimates between different layers. Please see the corresponding Figure R2 below.

Figure R2. Granger causality estimates between the untreated data (identical to the initial data presentation) compared with a modified data set with a 5-fold increased supragranular trace. We did not reveal any difference in the GC estimates which shows, that a mere amplitude increase may not explain the GC effects described in our study.

We also show the new figure 5 including caption for the reviewer’s direct inspection, as we have substantially revised it due to feedback from other reviewers:

Fig. 5. Analysis of time-domain pairwise-conditional granger causality. Granger causality matrices were calculated based on the single trial CSD traces from each cortical layer for each subject ($n=9$). **a** Median values of the granger causality estimates between cortical layers are shown before (*left*) and after (*right*) HYase treatment. The cortical column scheme depicts the dynamics of the predictive causal relationship across different cortical layers before and after ECM removal. Arrow thickness indicates the median of the causal estimate weights across animals. Red and green arrows indicate where G-causal forecasting undergoes a significant reduction or increase after HYase treatment, respectively. **b.** Comparison of the time-domain pairwise-conditional granger causality in the two conditions before and after HYase injection. *Left.* G-causal estimates were plotted with mean and standard error of the mean for each layer pairs from the 9 animals. *Right,* A multiple paired t-test was done on the log transformed values of the estimates and then corrected based on a false discovery rate of 0.05. Significant differences between G-causal measures indicated in the left plot correspond to a corrected level of significance after multiple comparisons of $p < 0.0078835$.

5) What does “weakened ECM” mean exactly (first paragraph of results)? In general, the manuscript would benefit from more precisely formulating the results and supporting them with quantitative analyses.

We agree with the reviewer, that the statement was rather vague. We used this term as alternative to ‘removal’, which is on the other hand a strong statement. To make this point more comprehensive, we now replaced the phrase ‘weakened’ with ‘significantly reduced’ and included a quantitative histological analysis of the ECM degradation as a new Figure 1d. We found a significant reduction of the *WFA* staining reflecting a reduction of the ECM. In contrast, and as a control, there was no changes of the PV+ staining of inhibitory interneurons. Please note, that we corrected the color channels for PV- and *WFA*-stainings compared to the initial submission (*WFA* antibody is coupled to fluorescein, and hence, should be labeled in green). For the reviewer’s inspection, we include Figure 1d and the corresponding caption in this response:

New Caption of Fig. 1 d Immunostainings after HYase treatment also revealed reduced density of *WFA* and hence reduced PNNs around PV+ interneurons. High-resolution confocal microscopy images (Zeiss LSM 700, Germany) show PV (red) and *WFA* (green) staining in the vicinity of the recording site. Stainings reveal layer-dependent accumulation of PNN structures in the auditory cortex. *Right*, Box plots represent the median (circle) and the interquartile range (25% and 75% percentile) and whiskers represent the full range of data (n = 4). Grey values of the *WFA* channel were significantly reduced for the HYase-treated side (Student’s t-Test; $p < 0.002$), but did not differ for PV+ ($p > 0.05$).

References

- Bikbaev, A., Frischknecht, R., and Heine, M. (2015). Brain extracellular matrix retains connectivity in neuronal networks. *Sci. Rep.* 5, 14527. doi:10.1038/srep14527.
- Brunk, M. G. K., Deane, K. E., Kisse, M., Deliano, M., Vieweg, S., Ohl, F. W., et al. (2019). Optogenetic stimulation of the VTA modulates a frequency-specific gain of thalamocortical inputs in infragranular layers of the auditory cortex. *Sci. Rep.* 9, https://doi.org/10.1038/s41598-019-56926-6. doi:10.1101/669168.
- de Vivo, L., Landi, S., Panniello, M., Baroncelli, L., Chierzi, S., Mariotti, L., et al. (2013). Extracellular matrix inhibits structural and functional plasticity of dendritic spines in the adult visual cortex. *Nat. Commun.* 4, 1484. doi:10.1038/ncomms2491.
- Deane, K. E., Brunk, M. G. K., Curran, A. W., Zempeltzi, M. M., Ma, J., Lin, X., et al. (2020). Ketamine anesthesia induces gain enhancement via recurrent excitation in granular input layers of the auditory cortex. *J. Physiol.* doi:10.1113/jp279705.
- Deliano, M., Brunk, M. G. K., El-Tabbal, M., Zempeltzi, M. M., Happel, M. F. K., and Ohl, F. W. (2018). Dopaminergic neuromodulation of high gamma stimulus phase-locking in gerbil primary auditory cortex mediated by D1/D5-receptors. *Eur. J. Neurosci.* doi:10.1111/ejn.13898.
- Happel, M. F. K., Deliano, M., Handschuh, J., and Ohl, F. W. (2014a). Dopamine-modulated recurrent corticoefferent feedback in primary sensory cortex promotes detection of behaviorally relevant stimuli. *J. Neurosci.* 34, 1234–47. doi:10.1523/JNEUROSCI.1990-13.2014.
- Happel, M. F. K., Jeschke, M., and Ohl, F. W. (2010). Spectral integration in primary auditory cortex attributable to temporally precise convergence of thalamocortical and intracortical input. *J. Neurosci.* 30, 11114–27. doi:10.1523/JNEUROSCI.0689-10.2010.
- Happel, M. F. K., Niekisch, H., Castiblanco Rivera, L. L., Ohl, F. W., Deliano, M., and Frischknecht, R. (2014b). Enhanced cognitive flexibility in reversal learning induced by removal of the

- extracellular matrix in auditory cortex. *Proc. Natl. Acad. Sci. U. S. A.* 111, 2800–2805. doi:10.1073/pnas.1310272111.
- Happel, M. F. K., and Ohl, F. W. (2017). Compensating Level-Dependent Frequency Representation in Auditory Cortex by Synaptic Integration of Corticocortical Input. *PLoS One* 12, e0169461. doi:10.1371/journal.pone.0169461.
- Jeschke, M., Happel, M. F. K., Tziridis, K., Krauss, P., Schilling, A., Schulze, H., et al. (2020). Acute and long-term circuit-level effects in the auditory cortex after sound trauma. *bioRxiv*.
- Ohl, F. W., and Scheich, H. (1997). Orderly cortical representation of vowels based on formant interaction. *Proc. Natl. Acad. Sci. U. S. A.* 94, 9440–9444. doi:10.1073/pnas.94.17.9440.
- Ohl, F. W., Wetzelsch, W., Wagner, T., Rech, A., and Scheich, H. (1999). Bilateral ablation of auditory cortex in Mongolian gerbil affects discrimination of frequency modulated tones but not of pure tones. *Learn. Mem.* 6, 347–362.
- Schaefer, M. K. K., Hechavarría, J. C. C., and Kössl, M. (2015). Quantification of mid and late evoked sinks in laminar current source density profiles of columns in the primary auditory cortex. *Front. Neural Circuits* 9, 1–16. doi:10.3389/fncir.2015.00052.
- Sherman, M. A., Lee, S., Law, R., Haegens, S., Thorn, C. A., Hämmäläinen, M. S., et al. (2016). Neural mechanisms of transient neocortical beta rhythms: Converging evidence from humans, computational modeling, monkeys, and mice. *Proc. Natl. Acad. Sci.* 113, E4885–E4894. doi:10.1073/pnas.1604135113.

REVIEWERS' COMMENTS:

Reviewer #1 (Remarks to the Author):

Revisions have addressed the reviewer's concerns and the paper is appropriate for publication in its present form.

Reviewer #2 (Remarks to the Author):

The manuscript is greatly improved, and the authors did a fantastic job addressing major concerns. In particular, the data provided in new Figure 1 and in the modified version of the Granger causality analyses, which are central to the conclusions of the paper, make the findings even more compelling.

Reviewer #3 (Remarks to the Author):

Thank you for the very detailed response and additional analyses/figures. I have more questions or concerns.